# HORIZON-FREE REGRET FOR LINEAR MARKOV DECISION PROCESSES

**Zihan Zhang**[*]  **Jason D. Lee**[*]  **Yuxin Chen**[†]  **Simon S. Du**[‡]

## ABSTRACT

A recent line of works showed regret bounds in reinforcement learning (RL) can be (nearly) independent of planning horizon, a.k.a. the horizon-free bounds. However, these regret bounds only apply to settings where a polynomial dependency on the size of transition model is allowed, such as tabular Markov Decision Process (MDP) and linear mixture MDP. We give the first horizon-free bound for the popular linear MDP setting where the size of the transition model can be exponentially large or even uncountable. In contrast to prior works which explicitly estimate the transition model and compute the inhomogeneous value functions at different time steps, we directly estimate the value functions and confidence sets. We obtain the horizon-free bound by: (1) maintaining *multiple* weighted least square estimators for the value functions; and (2) a structural lemma which shows the maximal total variation of the *inhomogeneous* value functions is bounded by a polynomial factor of the feature dimension.

## 1 INTRODUCTION

In reinforcement learning (RL), an agent learns to interact with an unknown environment by observing the current states and taking actions sequentially. The goal of the agent is to maximize the accumulative reward. In RL, the sample complexity describes the number of samples needed to learn a near-optimal policy.

It has been shown that the sample complexity needs to scale with the state-action space for the tabular Markov Decision Process (MDP) (Domingues et al., 2021). When the state-action space is large, function approximation is often used to generalize across states and actions. One popular model is linear MDP where the transition model is assumed to be low-rank (Yang & Wang, 2019; Jin et al., 2020b). We denote by $d$ the rank (or feature dimension) and the sample complexity will depend on $d$ instead of the size of the state-action space.

However, there remains a gap between our theoretical understanding of tabular MDPs and linear MDPs. For tabular MDPs, a line of works give horizon-free bounds, i.e., the sample complexity can be (nearly) independent of the planning horizon (Wang et al., 2020; Zhang et al., 2021a). More recently, the horizon-free bounds were also obtained in linear mixture MDP where the underlying MDP could be presented by a linear combination of $d$ known MDPs (Ayoub et al., 2020; Modi et al., 2020; Jia et al., 2020). A natural theoretical question is:

*Can we obtain horizon-free regret bounds for linear MDPs?*

One major technical challenge in obtaining horizon-free bounds is that the value function is inhomogeneous, i.e., for an MDP with planning horizon $H$, the optimal value functions for each time step $\{V_h^*\}_{h=1}^H$ can vary across $h = 1, \ldots, H$. For tabular MDPs and linear mixture MDPs, one can resolve this challenge by first estimating the transition kernel and then computing the value function based on the learned model. The sample complexity will then scale with the size of the transition

---
[*]Department of Electrical and Computer Engineering, Princeton University; email: `{zz5478,jasonlee}@princeton.edu`.

[†]Department of Statistics and Data Science, University of Pennsylvania; email: `yuxinc@wharton.upenn.edu`.

[‡]Paul G. Allen School of Computer Science and Engineering, University of Washington; email: `ssdu@cs.washington.edu`.

kernel, which is homogeneous and does not depend on $H$. For tabular MDPs and linear mixture MDPs, the dependence on the model size is allowed.[1] Unfortunately, this approach cannot be readily applied to handle linear MDPs because the model size of a linear MDP scales with the size of the state space, which can be exponentially large or even uncountable.

**Contributions.** In this paper, we answer the above question affirmatively by establishing a regret bound of $\widetilde{O}(\mathrm{poly}(d)\sqrt{K})$ for linear MDPs. Formally we have the result below.

**Theorem 1.** *Choose* `Reward − Confidence` *as VOFUL (see Algorithm 3). For any MDP satisfying the total-bounded reward assumption (Assumption 1) and linear MDP assumption (see Assumption 2), then with probability $1 − \delta$, the regret of Algorithm 1 is bounded by $\widetilde{O}(d^{5.5}\sqrt{K} + d^{6.5})$, where logarithmic factors of $(d, K, H, 1/\delta)$ are hidden by the $\widetilde{O}(\cdot)$ parameter.*

By virtue of Theorem 1, we show that linear MDP has a sample-complexity with only poly-logarithmic dependence on $H$. Although the proposed algorithm (Algorithm 1) is inefficient in computation, we believe our method provides intuitions to remove the horizon dependence in the view of statistical efficiency.

In terms of technical innovations, we design a novel method to share the samples to solve different linear bandit problems. Since the optimal value functions are not homogeneous, we need to learn $H$ different linear bandit problems using the same dataset. We first show that it suffices to bound the regret for each single bandit problem, and then bound the maximal total variance by bounding the variation of the optimal value function. See Section 3 for more details. Due to space limitation, we postpone the full proof of Theorem 1 to Appendix B.

## 1.1 RELATED WORKS

**Tabular MDPs.** There has been a long list of algorithms proposed for episodic tabular MDPs (e.g., Kearns & Singh (2002); Brafman & Tennenholtz (2003); Kakade (2003); Agrawal & Jia (2017); Azar et al. (2017); Jin et al. (2018); Zhang et al. (2020); Wang et al. (2020); Jin et al. (2020a); Zhang et al. (2021a); Li et al. (2021b;c; 2023); Zhang et al. (2022)). For finite-horizon *inhomogeneous* MDPs with the immediate reward at each step bounded by $1/H$, Azar et al. (2017); Zhang et al. (2020); Zanette & Brunskill (2019); Li et al. (2021b) achieved asymptotically minimax-optimal regret $\widetilde{\Theta}(\sqrt{SAHK})$ (ignoring lower order terms), where $SA$ is the size of state-action space, $H$ is the planning horizon, and $K$ is the number of episodes. Motivated by a conjecture raised by Jiang & Agarwal (2018), Wang et al. (2020) developed—for *time-homogeneous* MDPs with total rewards in any episode bounded above by 1—the first sample complexity upper bound that exhibits only logarithmic dependence on the horizon $H$, which was later on improved by Zhang et al. (2021a) to yield a near-optimal regret bound of $\widetilde{O}(\sqrt{SAK} + S^2A)$. Subsequently, Li et al. (2021c); Zhang et al. (2022) proved that even the poly-logarithmic horizon dependency in the sample complexity can be removed, albeit at the price of suboptimal scaling with $SA$.

**Contextual linear bandits.** The linear bandit problem has been extensively studied in past decades (Auer, 2002; Dani et al., 2008; Chu et al., 2011; Abbasi-Yadkori et al., 2011). For linear bandits with infinite arms, the minimax regret bound of $\widetilde{\Theta}(\sqrt{dK})$ is achieved by OFUL (Abbasi-Yadkori et al., 2011), where $d$ is the feature dimension and $K$ the number of rounds. With regards to variance-aware algorithms, Zhang et al. (2021b); Kim et al. (2022) proposed VOFUL (VOFUL+) to achieve a variance-dependent regret bound of $\widetilde{O}\big(\mathrm{poly}(d)\sqrt{\sum_{k=1}^{K}\sigma_k^2} + \mathrm{poly}(d)\big)$, in the absence of the knowledge of $\{\sigma_k\}_{k=1}^{K}$ (with $\sigma_k$ the conditional variance of the noise in the $k$-th round). By assuming prior knowledge of $\{\sigma_k\}_{k=1}^{K}$, Zhou et al. (2021); Zhou & Gu (2022) obtained similar regret bounds with improved dependence on $d$. Another work (Faury et al., 2020) proposed a Bernstein-style confidence set for the logistic bandit problem, also assuming availability of the noise variances.

**RL with linear function approximation.** It has been an important problem in the RL community to determine the generalization capability of linear function approximation (Jiang et al., 2017; Dann et al., 2018; Yang & Wang, 2019; Jin et al., 2020b; Wang et al., 2019; Sun et al., 2019; Zanette

---

[1] For linear mixture MDP, the model size scales linearly with the feature dimension.

et al., 2020; Weisz et al., 2020; Li et al., 2021a; Ayoub et al., 2020; Zhang et al., 2021b; Kim et al., 2022; Zhou et al., 2021; Zhou & Gu, 2022; He et al., 2022). Several model assumptions have been proposed and exploited to capture the underlying dynamics via linear functions. For example, Jiang et al. (2017) investigated low Bellman-rank, which described the algebraic dimension between the decision process and value-function approximator. Another setting proposed and studied by Jia et al. (2020); Ayoub et al. (2020); Modi et al. (2020) is that of linear mixture MDPs, which postulates that the underlying dynamics is a linear combination of $d$ known environments. Focusing on linear mixture MDPs, Zhang et al. (2021b) proposed the first sample-efficient algorithm to achieve horizon-free $\widetilde{O}(\text{poly}(d)\sqrt{K})$ regret, and later on Kim et al. (2022) obtained better $d$-dependency in the regret bound; further, a recent study Zhou & Gu (2022) designed a variance- & uncertainty-aware exploration bonus with weighted least-square regression, achieving near-optimal regret bounds with computation efficiency. Another recent strand of research Yang & Wang (2019); Jin et al. (2020b); He et al. (2022); Agarwal et al. (2022) studied the setting of linear MDPs, where the transition kernel and reward function are assumed to be linear functions of several known low-dimensional feature vectors. Take episodic inhomogeneous linear MDPs for example: when the feature dimension is $d$ and the immediate reward in each step is bounded above by $1/H$, (Jin et al., 2020b) established the regret bound of $\widetilde{O}(\sqrt{d^3 H^2 K})$, whereas the follow-up works He et al. (2022); Agarwal et al. (2022) improved the regret to $\widetilde{O}(d\sqrt{HK})$. It remained unclear whether and when horizon-free solutions are plausible in linear MDPs, in the hope of accommodating scenarios with exceedingly large $H$.

## 2 PRELIMINARIES

In this section, we present the basics of MDPs and the learning process, and introduce our key assumptions. Throughout the paper, $\Delta(X)$ denotes the set of probability distributions over the set $X$.

**Episodic MDPs.** A finite-horizon episodic MDP can be represented by a tuple $(\mathcal{S}, \mathcal{A}, R, P, K, H)$, where $\mathcal{S}$ denotes the state space containing $S$ states, $\mathcal{A}$ is the action space containing $A$ different actions, $R : \mathcal{S} \times \mathcal{A} \to \Delta([0,1])$ indicates the reward distribution, $P : \mathcal{S} \times \mathcal{A} \to \Delta(\mathcal{S})$ represents the probability transition kernel, $K$ stands for the total number of sample episodes that can be collected, and $H$ is the planning horizon. In particular, $P$ is assumed throughout to be *time-homogeneous*, which is necessary to enable nearly horizon-free regret bounds; in light of this assumption, we denote by $P_{s,a} := P(\cdot \mid s, a) \in \Delta(\mathcal{S})$ the transition probability from state $s$ to state $s'$ while taking action $a$. The reward distribution $R$ is also assumed to be time-homogeneous, so that the immediate reweard at a state-action pair $(s, a)$ at any step $h$ is drawn from $R(s, a)$ with mean $\mathbb{E}_{r' \sim R(s,a)}[r'] = r(s, a)$. Moreover, a deterministic and possibly non-stationary policy $\pi = \{\pi_h : \mathcal{S} \to \mathcal{A}\}_{h=1}^{H}$ describes an action selection strategy, with $\pi_h(s)$ specifying the action chosen in state $s$ at step $h$.

At each sample episode, the learner starts from an initial state $s_1$; for each step $h = 1, \ldots, H$, the learner observes the current state $s_h$, takes action $a_h$ accordingly, receives an immediate reward $r_h \sim R(s_h, a_h)$, and then the environment transits to the next state $s_{h+1}$ in accordance with $P(\cdot \mid s_h, a_h)$. When the actions are selected based on policy $\pi$, we can define the $Q$-function and the value function at step $h$ respectively as follows:

$$Q_h^\pi(s, a) := \mathbb{E}_\pi \left[ \sum_{h'=h}^{H} r_{h'} \,\middle|\, (s_h, a_h) = (s, a) \right] \quad \text{and} \quad V_h^\pi(s) := \mathbb{E}_\pi \left[ \sum_{h'=h}^{H} r_{h'} \,\middle|\, s_h = s \right]$$

for any $(s, a) \in \mathcal{S} \times \mathcal{A}$, where $\mathbb{E}_\pi[\cdot]$ denotes the expectation following $\pi$, i.e., we execute $a_{h'} = \pi_{h'}(s_{h'})$ for all $h < h' \leq H$ (resp. $h \leq h' \leq H$) in the definition of $Q_h^\pi$ (resp. $V_h^\pi$). The optimal $Q$-function and value function at step $h$ can then be defined respectively as

$$Q_h^*(s, a) = \max_\pi Q_h^\pi(s, a) \quad \text{and} \quad V_h^*(s) = \max_\pi V_h^\pi(s), \quad \forall (s, a) \in \mathcal{S} \times \mathcal{A}.$$

These functions satisfy the Bellman optimality condition in the sense that $V_h^*(s) = \max_a Q_h^*(s, a)$, $\forall s \in \mathcal{S}$, and $Q_h^*(s, a) = r(s, a) + \mathbb{E}_{s' \sim P(\cdot \mid s, a)}[V_{h+1}^*(s')], \forall (s, a) \in \mathcal{S} \times \mathcal{A}$.

**The learning process.** The learning process entails collection of $K$ sample episodes. At each episode $k = 1, 2, \ldots, K$, a policy $\pi^k$ is selected carefully based on the samples collected in the previous $k - 1$ episodes; the learner then starts from a given initial state $s_1^k$ and executes $\pi^k$ to collect

the $k$-th episode $\{(s_h^k, a_h^k, r_h^k)\}_{1 \le h \le H}$, where $s_h^k$, $a_h^k$ and $r_h^k$ denote respectively the state, action and immediate reward at step $h$ of this episode. The learning performance is measured by the total regret

$$\text{Regret}(K) := \sum_{k=1}^{K} \left( V_1^*(s_1^k) - V_1^{\pi^k}(s_1^k) \right), \tag{1}$$

and our ultimate goal is to design a learning algorithm that minimizes the above regret (1).

**Key assumptions.** We now introduce two key assumptions imposed throughout this paper, which play a crucial role in determining the minimal regret. The first assumption is imposed upon the rewards, requiring the aggregate reward in any episode to be bounded above by 1 almost surely.

**Assumption 1** (Bounded total rewards). *In any episode, we assume that $\sum_{h=1}^{H} r_h \le 1$ holds almost surely regardless of the policy in use.*

Compared to the common assumption where the immediate reward at each step is bounded by $1/H$, Assumption 1 is much weaker in that it allows the rewards to be spiky (e.g., we allow the immediate reward at one step to be on the order of 1 with the remaining ones being small). The interested reader is referred to Jiang & Agarwal (2018) for more discussions about the above reward assumption.

The second assumption postulates that the transition kernel and the reward function reside within some known low-dimensional subspace, a scenario commonly referred to as a linear MDP.

**Assumption 2** (Linear MDP (Jin et al., 2020b)). *Let $\mathcal{B}$ represent the unit $\ell_2$ ball in $\mathbb{R}^d$, and let $\{\phi(s,a)\}_{(s,a) \in \mathcal{S} \times \mathcal{A}} \subset \mathcal{B}$ be a set of known feature vectors such that $\max_{s,a} \|\phi_{s,a}\|_2 \le 1$. Assume that there exist a reward parameter $\theta_r \in \mathbb{R}^d$ and a transition kernel parameter $\mu \in \mathbb{R}^{S \times d}$ such that*

$$r(s,a) = \langle \phi(s,a), \theta_r \rangle \qquad \forall (s,a) \in \mathcal{S} \times \mathcal{A} \tag{2a}$$

$$P(\cdot \,|\, s, a) = \mu \phi(s, a), \qquad \forall (s, a) \in \mathcal{S} \times \mathcal{A} \tag{2b}$$

$$\|\theta_r\|_2 \le \sqrt{d}, \tag{2c}$$

$$\|\mu^\top v\|_2 \le \sqrt{d}, \qquad \forall v \in \mathbb{R}^S \text{ obeying } \|v\|_\infty \le 1. \tag{2d}$$

In words, Assumption 2 requires both the reward function and the transition kernel to be linear combinations of a set of $d$-dimensional feature vectors, which enables effective dimension reduction as long as $d \ll SA$.

In comparison, another line of works Jia et al. (2020); Ayoub et al. (2020); Modi et al. (2020) focus on the setting of linear mixture MDP below.

**Assumption 3** (Linear Mixture MDP). *Let $\{(r_i, P_i)\}_{i=1}^d$ be a group of known reward-transition pairs. Assume that there exists a kernel parameter $\theta \in \mathbb{R}^d$ such that*

$$r(s,a) = \sum_{i=1}^{d} \theta_i r_i(s,a) \qquad \forall (s,a) \in \mathcal{S} \times \mathcal{A} \tag{3a}$$

$$P(\cdot \,|\, s, a) = \sum_{i=1}^{d} \theta_i P_i(\cdot \,|\, s, a), \qquad \forall (s, a) \in \mathcal{S} \times \mathcal{A} \tag{3b}$$

$$\|\theta\|_1 \le 1. \tag{3c}$$

Roughly speaking, Assumption 3 requires that the underlying reward-transition pair is a linear combination of $d$ known reward-transition pairs. Recent work Zhou & Gu (2022) achieved a near-tight horizon-regret bound in this setting with a computational efficient algorithm. However, we emphasize that learning a linear MDP is fundamentally harder than learning a linear mixture MDP. The reason is that the only unknown parameter in a linear mixture MDP problem is the hidden kernel $\theta$, which has at most $d$ dimensions. So it is possible to learn $\theta$ to fully express the transition model. While in linear MDP, the dimension of unknown parameter $\mu$ scales linearly in the number of states, where it is impossible to recover the transition model. To address this problem, previous works on linear MDP try to learn the transition kernel in some certain direction, e.g., $\mu^\top v$ for some certain $v \in \mathbb{R}^S$. This approach faces a fundamental problem in sharing samples among difference layers. We refer to Section 3 for more discussion.

---

**Algorithm 1** Main Algorithm

---

1: **Input:** Number of episodes $K$, horizon $H$, feature dimension $d$, confidence parameter $\delta$

2: **Initialization:** $\lambda \leftarrow 1/H^2, \epsilon \leftarrow 1/(KH)^4, \alpha \leftarrow 150d\sqrt{\log^2((KH)/\delta)}$

3: **for** $k = 1, 2, \ldots, K$ **do**

4:    $\mathcal{D}^k \leftarrow \{s_{h'}^{k'}, a_{h'}^{k'}, s_{h'+1}^{k'}\}_{h' \in [H], k' \in [k-1]}$;

5:    *// Construct the confidence region for the transition kernel.*

6:    **for** $v \in \mathcal{W}_\epsilon$ **do**

7:       $(\hat{\theta}^k(v), \tilde{\theta}^k(v), \Lambda^k(v)) \leftarrow \mathtt{HF-Estimator}(\mathcal{D}^k, v)$;

8:       $b^k(v, \phi) \leftarrow \alpha\sqrt{\phi^\top(\Lambda^k(v))^{-1}\phi} + 4\epsilon$;

9:    **end for**

10:   $\mathcal{U}^k \leftarrow \left\{ \tilde{\mu} \in \mathcal{U} \,||\, \phi^\top\tilde{\mu}^\top v - \phi^\top\hat{\theta}(v)| \leq b^k(v, \phi), \forall \phi \in \Phi, v \in \mathcal{W}_\epsilon \right\}$

11:   *// Construct the confidence region for the reward function.*

12:   $\Theta^k \leftarrow \mathtt{Reward-Confidence}\left(\{\phi_h^{k'}/\sqrt{d}\}_{k' \in [k-1], h \in [H]}, \{r_h^{k'}/\sqrt{d}\}_{k' \in [k-1], h \in [H]}\right)$

13:   *// Optimistic planning.*

14:   $(\mu^k, \theta^k) \leftarrow \arg\max_{\tilde{\mu} \in \mathcal{U}^k, \theta \in \Theta^k} \max_\pi \mathbb{E}_\pi[\sum_{h=1}^H r_h | \tilde{\mu}, \theta]$;

15:   $\pi^k$ be the optimal policy w.r.t. the reward parameter as $\theta^k$ and transition parameter as $\mu^k$;

16:   Play $\pi^k$ in the $k$-th episode;

17: **end for**

---

**Algorithm 2** $\mathtt{HF-Estimator}$

---

1: **Input :** A group of samples $\mathcal{D} := \{s_i, a_i, s_i'\}_{i=1}^n$, value function $v \in \mathbb{R}^S$;

2: **Initialization:** $\lambda \leftarrow 1/H^2, \alpha \leftarrow 150d\sqrt{\log^2((KH)/\delta)}, \phi_i \leftarrow \phi(s_i, a_i), 1 \leq i \leq n, \epsilon \leftarrow 1/(KH)^4$;

3: $\sigma_1^2 \leftarrow 4$;

4: **for** $i = 2, 3, \ldots, n+1$ **do**

5:    $\Lambda_{i-1} \leftarrow \lambda\mathbf{I} + \sum_{i'=1}^{i-1} \phi_{i'}^\top\phi_{i'}/\sigma_{i'}^2$;

6:    $\tilde{b}_{i-1} \leftarrow \sum_{i'=1}^{i-1} \frac{v^2(s_{i'}')}{\sigma_{i'}^2}\phi_{i'}, \tilde{\theta}_{i-1} \leftarrow \Lambda_{i-1}^{-1}\tilde{b}_{i-1}$;

7:    $b_{i-1} \leftarrow \sum_{i'=1}^{i-1} \frac{v(s_{i'}')}{\sigma_{i'}^2}\phi_{i'}, \theta_{i-1} \leftarrow \Lambda_{i-1}^{-1}b_{i-1}$;

8:    $\sigma_i^2 \leftarrow \phi_i^\top\tilde{\theta}_{i-1} - (\phi_i^\top\theta_{i-1})^2 + 16\alpha\sqrt{\phi_i^\top(\Lambda_{i-1})^{-1}\phi_i} + 4\epsilon$,;

9: **end for**

10: $\theta \leftarrow \Lambda_n^{-1}b_n, \tilde{\theta} \leftarrow \Lambda_n^{-1}\tilde{b}_n, \Lambda \leftarrow \Lambda_n$;

11: **Return:** $(\theta, \tilde{\theta}, \Lambda)$;

---

**Notation.** Let us introduce several notation to be used throughout. First, we use $\iota$ to abbreviate $\log(2/\delta)$. For any $x \in \mathbb{R}^d$ and $\Lambda \in \mathbb{R}^{d \times d}$, we define the weighted norm $\|x\|_\Lambda := \sqrt{x^\top\Lambda x}$. Let $[N]$ denote the set $\{1, 2, \ldots, N\}$ for a positive integer $N$. Define $\mathcal{B}(x) := \{\theta \in \mathbb{R}^d \mid \|\theta\|_2 \leq x\}$ and let $\mathcal{B} := \mathcal{B}(1)$ be the unit ball. For two vectors $u, v$ with the same dimension, we say $u \geq v$ (resp. $u \leq v$) iff $u$ is elementwise no smaller (resp. larger) than $v$. For a random variable $X$, we use $\mathrm{Var}(X)$ to denote its variance. For any probability vector $p \in \Delta(\mathcal{S})$ and any $v = [v_i]_{1 \leq i \leq S} \in \mathbb{R}^S$, we denote by $\mathbb{V}(p, v) := p^\top(v^2) - (p^\top v)^2$ the associated variance, where $v^2 := [v_i^2]_{1 \leq i \leq S}$ denotes the entrywise square of $v$. Let $\phi_h^k$ abbreviate $\phi(s_h^k, a_h^k)$ for any proper $(h, k)$. Also, we say $(h', k') \leq (h, k)$ iff $h' + k'H \leq h + kH$. Let $\mathcal{F}_h^k$ denote the $\sigma$-algebra generated by $\{s_{h'}^{k'}, a_{h'}^{k'}\}_{(h',k') \leq (h,k)}$. We employ $\mathbb{E}[\cdot \mid \tilde{\mu}, \theta]$ to denote the expectation when the underlying linear MDP is generated by the transition kernel parameter $\tilde{\mu}$ and the reward parameter $\theta$ (cf. (2)). Moreover, let $\Phi$ denote the set of all possible features. Without loss of generality, we assume $\Phi$ is a convex set.

## 3 TECHNIQUE OVERVIEW

In this section, we first discuss the hardness of horizon-free bounds for linear MDP, and then introduce the high-level ideas of our approach. To simplify presentation, we focus on the regret incurred by learning the unknown transition dynamics.

### 3.1 TECHNICAL CHALLENGE

**Least-square regression in the linear MDP problem.** Jin et al. (2020c) proposed the first efficient algorithm (LSVI-UCB) for the linear MDP problem. In this method, for each $h \in [H]$, the learner maintains an estimation of $V_{h+1}$, and constructs optimistic estimators of $Q_h(s, a) := r(s, a) + P_{s,a}^\top V_{h+1}$. Since the reward $r$ is assumed to be known, it suffices to estimate $P_{s,a}^\top V_{h+1} = (\phi(s, a))^\top \mu^\top V_{h+1}$. By defining $\theta_{h+1} := \mu^\top V_{h+1}$, we can estimate $(\phi(s, a))^\top \theta_{h+1}$ with least-square regression because all state-action pairs share the same kernel $\theta_{h+1}$. This task is basically the same as a linear bandit problem, except for that additional factors are needed due to uniform bound over all possible choices $V_{h+1}$.

To obtain horizon-free regret bound, a common approach is to design estimators for $P_{s,a}^\top V_{h+1}$ with smaller confidence intervals. In this way, we can choose a smaller bonus to keep the optimism, and the regret is also reduced since the leading term in the regret is the sum of bonuses.

Recent work Zhou & Gu (2022) made progress in this direction by designing a variance-aware estimators for the linear regression problem. Roughly speaking, given a groups of samples $\{\phi_i, v_i\}_{i=1}^n$ such that (i) $v_i = \phi_i^\top \theta + \epsilon_i, \forall i \in [n]$; (ii) $\mathbb{E}[\epsilon_i | \{\phi_j\}_{j=1}^i, \{\epsilon_j\}_{j=1}^{i-1}] = 0$ and $\mathbb{E}[\epsilon_i^2 | \{\phi_j\}_{j=1}^i, \{\epsilon_j\}_{j=1}^{i-1}] = \sigma_i^2, \forall i \in [n]$, with the method in Zhou & Gu (2022), the width of the confidence interval of $\phi^\top \theta$ is roughly

$$\tilde{O}\left(\text{poly}(d)\sqrt{\phi^\top \Lambda^{-1}\phi}\right), \tag{4}$$

where $\Lambda = \lambda \mathbf{I} + \sum_{i=1}^n \frac{\phi_i \phi_i^\top}{\sigma_i^2}$ and $\lambda$ is some proper regularization parameter (See Lemma 1 in Appendix A).

**Main technical challenge: Variance-aware estimators coupled with inhomogeneous value functions.** While the transition kernel is assumed to be time-homogeneous, the value function and the policy can be *time-inhomogeneous* across different steps. Although the confidence width in (4) seems nice, it poses additional difficulty to bound the sum of bonuses due to *time-inhomogeneous* value functions.

Below we give more technical details to elucidate this technical challenge. To simplify the problem, we assume that the learner is informed of both the reward function and the optimal value function $\{V_h^*\}_{h \in [H]}$. Note that the arguments below can be extended to accommodate unknown $\{V_h^*\}_{h \in [H]}$ as well by means of proper exploitation of the linear MDP structure and a discretization method (i.e., applying a union bound over all possible optimal value functions over a suitably discretized set).

Let $\theta_h^* = \mu^\top V_{h+1}^*$. Then it reduces to learning $H$ contextual bandit problems with hidden parameter as $\{\theta_h^*\}_{h=1}^H$. To remove the polynomial factors of $H$, it is natural to share samples over different layers. That is, we need to use all the samples along the trajectory $\{s_{h'}, a_{h'}, s_{h'+1}\}_{h'=1}^H$ to estimate the value of $\phi^\top \theta_h^*$.

To solve the $h$-th linear bandit problem, following (4), we could get a regret bound of $\text{Regret}_h(K) := \tilde{O}\left(\sum_{k=1}^K \sqrt{(\phi_h^k)^\top (\Lambda^k(V_{h+1}^*))^{-1}\phi_h^k}\right)$. Here $\Lambda^k(v) = \lambda \mathbf{I} + \sum_{k'=1}^{k-1}\sum_{h'=1}^H \frac{\phi_{h'}^{k'}(\phi_{h'}^{k'})^\top}{(\sigma_{h'}^{k'}(v))^2}$ with $\left(\sigma_{h'}^{k'}(v)\right)^2$ as an upper bound for the variance $\mathbb{V}(P_{s_{h'}^{k'}, a_{h'}^{k'}}, v)$ for $v \in \mathbb{R}^S$. Taking sum over $h$, the resulting regret bound is roughly

$$\sum_{k=1}^K \min\left\{\sum_{h=1}^H \sqrt{(\phi_h^k)^\top (\Lambda^k(V_{h+1}^*))^{-1}\phi_h^k}, 1\right\}. \tag{5}$$

We remark that if $V_h^*$ is homogeneous in $h$, i.e., there exists $V^*$ such that $V_h^* = V^*$ for any $h \in [H]$, we could use Cauchy's inequality to bound (5) by[2]

$$\sqrt{\sum_{k=1}^{K} \min\left\{\sum_{h=1}^{H}(\phi_h^k/(\sigma_h^k(V^*)))^{\top}(\Lambda^k(V^*))^{-1}(\phi_h^k/\sigma_h^k(V^*)), 1\right\}} \cdot \sqrt{\sum_{k=1}^{K}\sum_{h=1}^{H}(\sigma_h^k(V^*))^2}. \quad (6)$$

Noting that

$$\Lambda^{k+1}(V^*) = \Lambda^k(V^*) + \sum_{h'=1}^{H} \frac{\phi_{h'}^k(\phi_{h'}^k)^{\top}}{\left(\sigma_{h'}^{k'}(V^*)\right)^2}, \quad (7)$$

we can further use the elliptical potential lemma (Lemma 6) to bound the first term in (6), and the total variance lemma for MDPs to bound the second term in (6). As a result, we can easily bound (5) by $\tilde{O}(\text{poly}(d)\sqrt{K})$.

However, the arguments above cannot work when $V_h^*$ depends on $h$. In such cases, the first term in (5) would be

$$\sqrt{\sum_{k=1}^{K} \min\left\{\sum_{h=1}^{H}(\phi_h^k/\sigma_h^k(V_h^*))^{\top}(\Lambda^k(V_{h+1}^*))^{-1}(\phi_h^k/\sigma_h^k(V_h^*)), 1\right\}}. \quad (8)$$

To invoke elliptical potential lemma, we need $\Lambda^{k+1}(V_{h+1}^*) - \Lambda^k(V_{h+1}^*) = \sum_{h'=1}^{H} \frac{\phi_{h'}^k(\phi_{h'}^k)^{\top}}{(\sigma_{h'}^k(V_{h'+1}^*))^2}$, which does not hold since $\sigma_{h'}^k(V_{h'+1}^*) \neq \sigma_{h'}^k(V_{h+1}^*)$.

In comparison, for tabular MDP, the variance aware bonus has a simple form of $\sqrt{\frac{\mathbb{V}(P_{s,a}, V_{h+1}^*)}{N}}$, so that one can invoke Cauchy's inequality to bound the sum of bonuses; for linear mixture MDP, because there is only one kernel parameter $\theta$ and one information matrix, it suffices to analyze like (6) and (7).

## 3.2 OUR METHODS

In high-level idea, by noting that the main obstacle is the *time-inhomogeneous* value function, we aim to prove that the value function $\{V_h^*\}_{h=1}^{H}$ could be divided into several groups such that in each group, the value functions are similar to each other measured by the variance.

**Technique 1: a uniform upper bound for the variances.** We consider using a uniform upper bound $(\bar{\sigma}_{h'}^{k'})^2 := \max_{h \in [H]}(\sigma_{h'}^{k'}(V_{h+1}^*))^2$ to replace $(\sigma_{h'}^{k'}(V_{h+1}^*))^2$ when computing $\Lambda^k(V_{h+1}^*)$. That is, by setting $\bar{\Lambda}^k = \lambda \mathbf{I} + \sum_{k'=1}^{k-1}\sum_{h'=1}^{H} \frac{\phi_{h'}^{k'}(\phi_{h'}^{k'})^{\top}}{(\bar{\sigma}_{h'}^{k'})^2} \preccurlyeq \Lambda^k(V_{h+1}^*)$ for any $h \in [H]$, we can bound (5) as below:

$$\sum_{k=1}^{K} \min\left\{\sum_{h=1}^{H}\sqrt{(\phi_h^k)^{\top}(\Lambda^k(V_{h+1}^*))^{-1}\phi_h^k}, 1\right\}$$

$$\leq \sum_{k=1}^{K} \min\left\{\sum_{h=1}^{H}\sqrt{(\phi_h^k)^{\top}(\bar{\Lambda}^k)^{-1}\phi_h^k}, 1\right\}$$

$$\approx \sqrt{\sum_{k=1}^{K} \min\left\{\sum_{h=1}^{H}(\phi_h^k/\bar{\sigma}_h^k)^{\top}(\bar{\Lambda}^k)^{-1}(\phi_h^k/\bar{\sigma}_h^k), 1\right\}} \cdot \sqrt{\sum_{k=1}^{K}\sum_{h=1}^{H}(\bar{\sigma}_h^k)^2}. \quad (9)$$

With the elliptical potential lemma (Lemma 6), we have that

$$\sum_{k=1}^{K} \min\left\{\sum_{h=1}^{H}(\phi_h^k/\bar{\sigma}_h^k)^{\top}(\bar{\Lambda}^k)^{-1}(\phi_h^k/\bar{\sigma}_h^k), 1\right\} = \tilde{O}(\sqrt{d}).$$

---

[2]Here we omit a lower order term.

So it suffices to deal with $\sum_{k=1}^{K} \sum_{h=1}^{H} (\bar{\sigma}_h^k)^2$. For simplicity, we assume that $(\sigma_h^k(v))^2$ is exactly $\mathbb{V}(P_{s_h^k, a_h^k}, v)$ and consider to bound $\sum_{k=1}^{K} \sum_{h=1}^{H} \max_{h' \in [H]} \mathbb{V}(P_{s_h^k, a_h^k}, V_{h'+1}^*)$.

Noting that $\mathbb{V}(P_{s,a}, v)$ could be written as $\phi(s,a)^\top(\theta(v^2)) - (\phi(s,a)^\top \theta(v))^2$, which is a linear function of the matrix $\begin{bmatrix} \phi(s,a)\phi^\top(s,a) & \phi(s,a) \\ \phi^\top(s,a) & 1 \end{bmatrix}$, we can bound $\sum_{k=1}^{K} \sum_{h=1}^{H} \max_{h' \in [H]} \mathbb{V}(P_{s_h^k, a_h^k}, V_{h'+1}^*)$ by $2(d+1)^2 \max_{h' \in [H]} \sum_{k=1}^{K} \sum_{h=1}^{H} \mathbb{V}(P_{s_h^k, a_h^k}, V_{h'+1}^*)$ with a useful technical lemma (See Lemma 5.)

As a result, it suffices to bound $\sum_{k=1}^{K} \sum_{h=1}^{H} \mathbb{V}(P_{s_h^k, a_h^k}, V_{h'+1}^*)$ for each $h' \in [H]$. However, because $V_{h'+1}^*$ can vary significantly when $h'$ is closed to $H$, $\sum_{k=1}^{K} \sum_{h=1}^{H} \mathbb{V}(P_{s_h^k, a_h^k}, V_{h'+1}^*)$ might be large in the worst case. We consider the toy example below.

**Example 1.** *Fix some $\epsilon > 0$. Let $\mathcal{S} := \{s_1, s_2, s_3, z\}$, $\mathcal{A} = \{a_1, a_2\}$. Let $P_{s_1, a_1} = P_{s_2, a_1} = [\frac{1}{2} - \epsilon, \frac{1}{2} - \epsilon, \epsilon, 0]^\top$, $r(s_1, a_1) = r(s_2, a_1) = 0$, $P_{s_1, a_2} = P_{s_2, a_2} = [0, 0, 0, 1]^\top$, $r(s_1, a_2) = \frac{1}{2}$, $r(s_2, a_2) = 0$, $P_{s_3, a_1} = P_{s_3, a_2} = [0, 0, 0, 1]^\top$, $r(s_3, a_1) = r(s_3, a_2) = 1$, $P_{z, a_1} = P_{z, a_2} = [0, 0, 0, 1]^\top$, and $r(z, a_1) = r(z, a_2) = 0$.*

In this toy example, we have two frequent states $\{s_1, s_2\}$, one transient state $\{s_3\}$ with reward 1 and one ending state $z$ with no reward. The transition dynamics at $\{s_1, s_2\}$ is the same, but one can get reward $\frac{1}{2}$ in one step by taking action $a_2$ at $s_1$. Suppose $H >> \frac{1}{\epsilon}$ and $h \le \frac{H}{2}$, then the optimal action for $\{s_1, s_2\}$ at the $h$-th step should be $a_1$, and $V_h^*(s_1) \approx V_h^*(s_2) \approx 1$. On the other hand, it is easy to observe $V_H^*(s_1) = \frac{1}{2}$ and $V_H^*(s_2) = 0$. Let the initial state be $s_1$. Following the optimal policy, we have $\mathbb{E}\left[\sum_{h=1}^{\frac{H}{2}} \mathbb{V}(P_{s_h^k, a_h^k}, V_H^*)\right] = \Omega(\frac{1}{\epsilon}) >> 1$ when choosing $\epsilon$ small enough.

**Technique 2: bounding the total variation.** Direct computation shows that for $1 \le h_1 < h_2 \le [H]$,

$$\sum_{k=1}^{K} \sum_{h=h_1}^{h_2} \mathbb{V}(P_{s_h^k, a_h^k}, V_{h'+1}^*) = \tilde{O}(K + K(h_2 - h_1 + 1)\|V_{h'}^* - V_{h'+1}^*\|_\infty). \tag{10}$$

Let $l_h = \|V_h^* - V_{h+1}^*\|_\infty$. It is easy to observe that $l_h \le l_{h+1}$ for $1 \le h \le H - 1$ since the Bellman operator $\Gamma$ is a contraction , i.e., $\|\Gamma(v_1 - v_2)\|_\infty \le \|v_1 - v_2\|_\infty$ for any $v_1, v_2 \in \mathbb{R}^S$. So we can obtain $l_h \le \frac{\sum_{h'=1}^{H-1} l_{h'}}{H-h+1}$. For tabular MDP, it is easy to bound $\sum_{h=1}^{H} l_h \le S$ since $\|V_h^* - V_{h+1}^*\|_\infty \le \sum_s (V_h(s) - V_{h+1}(s))$. As a generalization to linear MDP, by Lemma 5 we have that

$$\sum_{h=1}^{H-1} l_h^* \le \sum_{h=1}^{H-1} \max_{\phi \in \Phi} \phi^\top \mu^\top (V_{h+1} - V_{h+2}) \le \max_{\phi \in \Phi} 2d\phi^\top \sum_{h=1}^{H-1} \mu^\top (V_{h+1} - V_{h+2}) \le 2d. \tag{11}$$

As a result, $l_h^* \le \frac{2d}{H-h+1}$.

**Technique 3: doubling segments.** By choosing $h_1 = \frac{H}{2} + 1$ and $h_2 = H$ in (10), for $h' \in [h_1, h_2]$,

$$\sum_{k=1}^{K} \sum_{h=h_1}^{h_2} \mathbb{V}(P_{s_h^k, a_h^k}, V_{h'+1}^*) = \tilde{O}(K + K(h_2 - h_1 + 1)\|V_{h'}^* - V_{h'+1}^*\|_\infty) = \tilde{O}(Kd).$$

This inspires us to divide $[H]$ several segments $[H] = \cup_i \mathcal{H}_i$ with $\mathcal{H}_i = \{h | H - \frac{H}{2^{i-1}} + 1 \le h \le H - \frac{H}{2^i}\}$ and $\mathcal{H}_{\log_2(H)+1} = \{H\}$[3]. Consequently, for any $i$ and $h' \in \mathcal{H}_i$, using (10) and the fact that $l_{h'}^* \le \frac{2d}{H-h'+1} \le \frac{2^{i+1}d}{H}$, $\sum_{k=1}^{K} \sum_{h \in \mathcal{H}_i} \mathbb{V}(P_{s_h^k, a_h^k}, V_{h'+1}^*) = \tilde{O}(Kd)$.

Note that we only bound $\sum_{k=1}^{K} \sum_{i=1}^{\log_2(H)+1} \max_{h' \in \mathcal{H}_i} \sum_{h \in \mathcal{H}_i} \mathbb{V}(P_{s_h^k, a_h^k}, V_{h'+1}^*)$, which does not imply any bound for $\max_{h' \in [H]} \sum_{k=1}^{K} \sum_{h=1}^{H} \mathbb{V}(P_{s_h^k, a_h^k}, V_{h'+1}^*)$. Recall that our initial target is to

---

[3]We assume $\log_2(H)$ is an integer without loss of generality.

bound $\sum_{k=1}^{K} \min \left\{ \sum_{h=1}^{H} \sqrt{(\phi_h^k)^\top (\Lambda^k(V_{h+1}^*))^{-1} \phi_h^k}, 1 \right\}$. A natural idea is to group $h \in \mathcal{H}_i$ for each $i$ to avoid the term $\max_{h' \in [H]} \sum_{k=1}^{K} \sum_{h=1}^{H} \mathbb{V}(P_{s_h^k, a_h^k}, V_{h'+1}^*)$. In other words, we turn to bound $\sum_{k=1}^{K} \min \left\{ \sum_{h \in \mathcal{H}_i} \sqrt{(\phi_h^k)^\top (\Lambda^k(V_{h+1}^*))^{-1} \phi_h^k}, 1 \right\}$ for each $i$ separately. More precisely, for fixed $i$, we let $(\bar{\sigma}_{h'}^{k'})^2 = \max_{h \in \mathcal{H}_i} (\sigma_{h'}^{k'}(V_{h+1}^*))^2$, and $\bar{\Lambda}^k = \lambda \mathbf{I} + \sum_{k'=1}^{k-1} \sum_{h' \in \mathcal{H}_i} \frac{\phi_{h'}^{k'}(\phi_{h'}^{k'})^\top}{(\bar{\sigma}_{h'}^{k'})^2}$. With the arguments above, we have that

$$
\begin{aligned}
&\sum_{k=1}^{K} \min \left\{ \sum_{h \in \mathcal{H}_i} \sqrt{(\phi_h^k)^\top (\Lambda^k(V_{h+1}^*))^{-1} \phi_h^k}, 1 \right\} \\
&\leq \sqrt{\sum_{k=1}^{K} \min \left\{ \sum_{h \in \mathcal{H}_i} (\phi_h^k / \bar{\sigma}_h^k)(\bar{\Lambda}^k)^{-1}(\phi_h^k / \bar{\sigma}_h^k), 1 \right\}} \cdot \sqrt{\sum_{k=1}^{K} \sum_{h \in \mathcal{H}_i} (\bar{\sigma}_h^k)^2 + \tilde{O}(d)} \\
&= \tilde{O}(\sqrt{K d^4}).
\end{aligned}
\tag{12}
$$

## 4 ALGORITHM

In this section, we introduce Algorithm 1. The algorithm is based on model-elimination. At each episode $k = 1, 2, \ldots, K$, we maintain $\mathcal{U}^k$ as confidence region of $\mu$ and $\Theta^k$ as confidence region for $\theta_r$. Then we select the optimistic transition model and reward function from $\mathcal{U}^k \times \Theta^k$ and then execute the corresponding optimal policy. The key step is how to construct $\mathcal{U}^k$. Inspired by recent work Zhou & Gu (2022), we consider the weighted least square regression to estimate the value function and corresponding variance, which is presented in Algorithm 2. We also borrow VOFUL in Zhang et al. (2021b) to construct the confidence region for $\theta_r$.

Recall that $\mathcal{B}(2\sqrt{d}) = \{\theta \in \mathbb{R}^d | \|\theta\|_2 \leq 2\sqrt{d}\}$. For fixed $\epsilon > 0$, there exists an $\epsilon$-net $\mathcal{B}_\epsilon(2\sqrt{d})$ w.r.t. $L_\infty$ for $\mathcal{B}(2\sqrt{d})$ such that $|\mathcal{B}_\epsilon(2\sqrt{d})| \leq O((4\sqrt{d}/\epsilon)^d)$. By Assumption 2, for any $v \in \mathbb{R}^S$ such that $\|v\|_\infty \leq 1$, it holds that $\|\mu^\top v\|_2 \leq \sqrt{d}$. Therefore, for any MDP such that Assumption 1 and 2 holds, its optimal value function is in the set

$$
\mathcal{W} := \{v \in \mathbb{R}^S | \exists \theta \in \mathcal{B}(2\sqrt{d}), v(s) = \max\{\min\{\max_a \phi^\top(s,a)\theta, 1\}, 0\}, \forall s \in \mathcal{S}\}.
$$

Define $\mathcal{W}_\epsilon = \left\{ v \in \mathbb{R}^S | \exists \theta \in \mathcal{B}_\epsilon(2\sqrt{d}), v(s) = \max\{\min\{\max_a \phi^\top(s,a)\theta, 1\}, 0\}, \forall s \in \mathcal{S} \right\}$. For fixed $\theta \in \mathcal{B}(2\sqrt{d})$ and $s \in \mathcal{S}$, the function $\max\left\{\min\left\{\max_a \phi(s,a)^\top \theta, 1\right\}, 0\right\}$ is $O(1)$-Lipschtiz continuous w.r.t $L_\infty$ norm. As a result, $\mathcal{W}_\epsilon$ is an $\epsilon$-net w.r.t. $L_\infty$ norm of $\mathcal{W}$. Besides, the size of $\mathcal{W}_\epsilon$ is bounded by $|\mathcal{W}_\epsilon| = O((4\sqrt{d}/\epsilon)^d)$.

**Confidence region for the transition kernel.** Fix a group of sequential samples $\{\phi_i\}_{i=1}^n$ and a value function $v \in \mathcal{W}_\epsilon$. Fix $\phi \in \Phi$ and let $\theta(v) = \mu^\top v$. We aim to construct a confidence interval from $\phi^\top \mu^\top v$, and then eliminate all the transition kernels $\tilde{\mu}$ which fails to satisfy the confidence interval for some $v$ and $\phi$. To obtain variance-aware confidence interval, we need to compute the variance to feed the weight least-square estimator in Zhou & Gu (2022). For this purpose, for the $i$-th variance $\mathbb{V}(\mu\phi_i, v)$, we construct $\sigma_i^2$ such that $\sigma_i^2 \geq \mathbb{V}(\mu\phi_i, v)$ and the error $\sigma_i^2 - \mathbb{V}(\mu\phi_i, v)$ is well controlled. To compute $\sigma_i^2$, we need to estimate $\phi_i^\top \theta(v^2)$ and $\phi_i^\top \theta(v)$ using the first $i-1$-th samples, which requires the knowledge of $\mathbb{V}(\mu\phi_{i'}, v^2)$ for $i' \leq i - 1$. To address this problem, Zhou & Gu (2022) recursively estimated the $2^m$-th order momentum for $m = 1, 2, \ldots, \log_2(H)$. In comparison, by the fact that $\mathbb{V}(\mu\phi_{i'}, v^2) \leq 4\mathbb{V}(\mu\phi_{i'}, v)$ (see Lemma 2), we can use $4\sigma_{i'}^2$ as an upper bound for $\mathbb{V}(\mu\phi_{i'}, v^2)$.

**Confidence region for the reward parameter.** To estimate the reward parameter $\theta_r$, we invoke VOFUL in Zhang et al. (2021b). We remark that the randomness in reward is independent of the randomness in transition dynamics, so that learning the transition dynamic does not help to estimate the variance of reward. More precisely, the variance of $R(s,a)$ could be a non-linear function in $\phi(s,a)$, while the variance of $V_h^*(s')$ with $s' \sim P(\cdot|s,a)$ must be a linear function in $\phi(s,a)$. In Appendix C, we present VOFUL and summarize some useful properties to bound the error due to uncertainty of reward parameter.

ACKNOWLEDGEMENT

Y. Chen is supported in part by the Alfred P. Sloan Research Fellowship, the Google Research Scholar Award, the AFOSR grants FA9550-19-1-0030 and FA9550-22-1-0198, the ONR grant N00014-22-1-2354, and the NSF grants CCF-2221009 and CCF-1907661. JDL acknowledges support of the ARO under MURI Award W911NF-11-1-0304, the Sloan Research Fellowship, NSF CCF 2002272, NSF IIS 2107304, NSF CIF 2212262, ONR Young Investigator Award, and NSF CAREER Award 2144994. SSD acknowledges the support of NSF IIS 2110170, NSFDMS 2134106, NSF CCF 2212261, NSF IIS 2143493, NSF CCF 2019844, NSF IIS 2229881.

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
