## A TECHNICAL LEMMAS

**Lemma 1** (Theorem 4.3 in Zhou & Gu (2022)). *Let $\{\mathcal{F}_i\}_{i=1}^{\infty}$ be a filtration, and $\{\psi_i, \zeta_i\}_{i \geq 1}$ be a stochastic process such that $\psi_i \in \mathbb{R}^d$ is $\mathcal{F}_i$-measurable and $\epsilon_i \in \mathbb{R}$ is $\mathcal{F}_{i+1}$ measurable. Let $L, \sigma, R, \lambda, c > 0, \theta^* \in \mathbb{R}^d$. For $i \geq 1$, let $y_i = \langle \psi_i, \theta^* \rangle + \zeta_i$ and suppose that*

$$\mathbb{E}[\zeta_i | \mathcal{F}_i] = 0, \quad \mathbb{E}[\zeta_i^2 | \mathcal{F}_i] \leq \sigma^2, |\zeta_i| \leq R, \|\psi_i\|_2 \leq L. \tag{13}$$

*For $i \geq 1$, let $\Lambda_i = \lambda \mathbf{I} + \sum_{i'=1}^{i} \psi_{i'} \psi_{i'}^{\top}$ and $b_i = \sum_{i'=1}^{i} y_{i'} \psi_{i'}$, $\theta_i = \Lambda_i^{-1} b_i$, and*

$$\kappa_i = 12\sqrt{\sigma^2 d \log(1 + iL^2/(d\lambda)) \log(32(\log(R/c) + 1)i^2/\delta)}$$
$$+ 24 \log(32(\log(R/c) + 1)k^2/\delta) \max_{1 \leq i' \leq i}\{|\zeta_{i'}| \min\{1, \|\psi_{i'}\|_{\Lambda_{i'-1}^{-1}}\}\} + 6 \log(32(\log(R/c) + 1)k^2/\delta)c.$$

*Then, for any $0 < \delta < 1$, with probability $1 - \delta$.*

$$\forall i \geq 1, \quad \left\| \sum_{i'=1}^{i} \psi_{i'} \zeta_{i'} \right\|_{\Lambda_i^{-1}} \leq \kappa_i, \quad \|\theta_i - \theta^*\|_{\Lambda_i} \leq \kappa_i + \sqrt{\lambda}\|\theta^*\|_2.$$

**Lemma 2** ( Chen et al. (2021)). *Let $X$ be a random variable taking value in $[-C, C]$ for some $C \geq 0$. Let $\mathrm{var}(Y)$ denote the variance of a random variable $Y$. It then holds that $\mathrm{var}(X^2) \leq 4C^2 \mathrm{var}(X)$.*

**Lemma 3.** *[Lemma 10 in Zhang et al. (2020)] Let $(M_n)_{n \geq 0}$ be a martingale such that $M_0 = 0$ and $|M_n - M_{n-1}| \leq c$ for some $c > 0$ and any $n \geq 1$. Let $\mathrm{Var}_n = \sum_{k=1}^{n} \mathbb{E}[(M_k - M_{k-1})^2 | \mathcal{F}_{k-1}]$ for $n \geq 0$, where $\mathcal{F}_k = \sigma(M_1, M_2, ..., M_k)$. Then for any positive integer $n$, and any $\epsilon, p > 0$, we have that*

$$\mathbb{P}\left[|M_n| \geq 2\sqrt{\mathrm{Var}_n \log(\frac{1}{p})} + 2\sqrt{\epsilon \log(\frac{1}{p})} + 2c \log(\frac{1}{p})\right] \leq \left(\frac{2nc^2}{\epsilon} + 2\right)p. \tag{14}$$

**Lemma 4.** *Let $\Phi$ be a bounded convex closed subset of $\mathbb{R}^d$. Let $\Phi^* = \{\psi \in \mathbb{R}^d | \phi^{\top} \psi \geq 0, \forall \phi \in \Phi\}$. Then there exists $\bar{\phi} \in \Phi$ satisfying that $\bar{\phi}^{\top} \psi \geq \frac{1}{2d} \max_{\phi \in \Phi} \phi^{\top} \psi$ for any $\psi \in \Phi^*$.*

*Proof.* Without loss of generality, we assume that $\|\psi\|_2 = 1$. Let $u$ be the standard measure of $\mathbb{R}^d$. Define $\bar{\phi} = \frac{\int_{\Phi} \phi du(\phi)}{\int_{\Phi} du(\phi)}$ be the geometry center of $\Phi$. We will show that $\bar{\phi}^{\top} \psi \geq \frac{1}{2d} \max_{\phi \in \Phi} \phi^{\top} \psi$ for any $\psi \in \Phi^*$. Fix $\psi \in \Phi^*$ and define $l = \max_{\phi \in \Phi} \phi^{\top} \psi$. For $0 \leq x \leq l$, we define $f(x) = \lim_{\epsilon \to 0} \frac{\int_{\phi \in \Phi, \phi^{\top} \psi \in [x, x+\epsilon]} du(\phi)}{\epsilon \cdot \int_{\phi} du(\phi)}$. Because $\Phi$ is a convex bounded set, $f(x)$ is well defined and continuous. By definition, it holds that $\int_0^l f(x)dx = 1$.

We claim that

$$\frac{f(l-y)}{y^{d-1}} \leq \frac{f(l-x)}{x^{d-1}} \tag{15}$$

for $0 < x \leq y \leq l$.

**Proof of** (15). Let $\mathcal{V}(w) := \{\phi \in \Phi : \phi^{\top} \psi = w\}$ for $0 \leq w \leq l$. $\mathcal{V}(l)$ is non-empty by definition. Fix $\tilde{\phi} \in \mathcal{V}(l)$. By convexity of $\Phi$, for any $\phi \in \mathcal{V}(l-y)$, $\frac{y-x}{y}\tilde{\phi} + \frac{x}{y}\phi \in \mathcal{V}(l-x)$ for any $0 < x \leq y \leq l$. As a result,

$$\frac{y-x}{y}\tilde{\phi} + \frac{x}{y}\mathcal{V}(l-y) := \left\{ \frac{y-x}{y}\tilde{\phi} + \frac{x}{y}\phi \mid \phi \in \mathcal{V}(l-y) \right\} \subset \mathcal{V}(l-x).$$

Let $\tilde{u}$ be the standard measure of $\mathbb{R}^{d-1}$. It then holds that

$$\tilde{u}(\mathcal{V}(l-x)) \geq \tilde{u}(\mathcal{V}(l-y)) \cdot \left(\frac{x}{y}\right)^{d-1} \tag{16}$$

for $0 < x \le y \le l$. By definition of $f$, we have that

$$\lim_{\epsilon \to 0} \frac{\epsilon \min_{w' \in [w, w+\epsilon]} \tilde{u}(\mathcal{V}(w'))}{\epsilon \int_{\Phi} du(\phi)}$$

$$\le f(w) := \lim_{\epsilon \to 0} \frac{\|\psi\|_2 \int_{\phi \in \Phi, \phi^\top \psi \in [w, w+\epsilon]} du(\phi)}{\epsilon \int_{\Phi} du(\phi)}$$

$$= \lim_{\epsilon \to 0} \frac{\epsilon \max_{w' \in [w, w+\epsilon]} \tilde{u}(\mathcal{V}(w'))}{\epsilon \int_{\Phi} du(\phi)}.$$

In then follows that

$$\lim_{\epsilon \to 0} \frac{\min_{w' \in [w, w+\epsilon]} \tilde{u}(\mathcal{V}(w'))}{\int_{\Phi} du(\phi)} \le f(w) \le \lim_{\epsilon \to 0} \frac{\max_{w' \in [w, w+\epsilon]} \tilde{u}(\mathcal{V}(w'))}{\int_{\Phi} du(\phi)}.$$

As a result, we have that

$$\frac{f(l-y)}{y^{d-1}} \le \lim_{\epsilon \to 0} \max_{v \in [0, \epsilon)]} \frac{\tilde{u}(\mathcal{V}(l-y+v))}{y^{d-1} \int_{\Phi} du(\phi)}$$

$$\le \lim_{\epsilon \to 0} \max_{v \in [0, \epsilon)} \min_{v' \in [0, \epsilon)} \frac{\tilde{u}(\mathcal{V}(l-x+v'))}{x^{d-1} \int_{\Phi} du(\phi)} \cdot \left( \frac{(y-v)x}{y(x-v')} \right)^{d-1}$$

$$\le \frac{f(l-x)}{x^{d-1}}.$$

The proof of (15) is finished.

Let $z = \bar{\phi}^\top \psi$ and $f(z) = b$, we then have that $f(x) \ge \frac{b \cdot (l-x)^{d-1}}{(l-z)^{d-1}}$ for any $x \in [z, l]$, and $f(x) \le \frac{b(l-x)^{d-1}}{(l-z)^{d-1}}$ for any $x \in [z]$.

By definition, we have that

$$z = \frac{\int_{\Phi} \phi^\top \psi du(\phi)}{\int_{\phi} du(\phi)}$$

$$= \lim_{\epsilon \to 0} \sum_{i=0}^{\lceil l/\epsilon \rceil} \frac{\int_{\phi \in \Phi, \phi^\top \psi \in [i\epsilon, (i+1)\epsilon)} \phi^\top \psi du(\phi)}{\int_{\Phi} du(\phi)}$$

$$= \lim_{\epsilon \to 0} \sum_{i=0}^{\lceil l/\epsilon \rceil} \frac{\int_{\phi \in \Phi, \phi^\top \psi \in [i\epsilon, (i+1)\epsilon)} i\epsilon du(\phi)}{\int_{\Phi} du(\phi)}$$

$$= \lim_{\epsilon \to 0} \sum_{i=0}^{\lceil l/\epsilon \rceil} f(i\epsilon) i\epsilon^2$$

$$= \int_0^l f(x) x dx.$$

As a result, we have that

$$\int_{0 \le x \le z} f(x) \cdot (z-x) d(x) = \int_{z \le x \le l} f(x)(x-z) dx,$$

which implies

$$\int_{z \le x \le l} \frac{b(l-x)^{d-1}(x-z)}{(l-z)^{d-1}} \le \int_{z \le x \le l} f(x)(x-z) dx = \int_{0 \le x \le z} f(x) \cdot (z-x) d(x) \le \int_{0 \le x \le z} \frac{b(l-x)^{d-1}(z-x)}{(l-z)^{d-1}}.$$

Therefore, $\int_{z \le x \le l} (l-x)^{d-1}(x-z) dx \le \int_{0 \le x \le z} (l-x)^{d-1}(z-x) dx$, which means

$$\frac{(l-z)^{d+1}}{d(d+1)} \le \frac{l^{d+1}}{d+1} - \frac{(l-z)l^d}{d} + \frac{(l-z)^{d+1}}{d(d+1)}. \tag{17}$$

Then it holds that $z \ge \frac{l}{d+1} \ge \frac{l}{2d}$. The proof is completed. $\square$

**Lemma 5.** *Let $l$ be a positive integer. Let $\Phi = \{\phi_i\}_{i=1}^n$ and $\Psi = \{\psi_j\}_{j=1}^m$ be two group of vectors in $\mathbb{R}^l$ satisfying that $\phi_i^\top \psi_j \geq 0$ for any $1 \leq i \leq n$ and $1 \leq j \leq m$. It then holds that*

$$\sum_{i=1}^n \max_j \phi_i^\top \psi_j \leq 2l \max_j \sum_{i=1}^n \phi_i^\top \psi_j.$$

*Proof.* By Lemma 4, there exists $\psi^* \in \text{Conv}(\Psi)$ such that $\phi_i^\top \psi^* \geq \frac{1}{2l} \max_j \phi_i^\top \psi_j$ for any $1 \leq i \leq n$. As a result, we have that

$$\sum_{i=1}^n \max_j \phi_i^\top \psi_j \leq 2l \sum_{i=1}^n \phi_i^\top \psi^* \leq 2l \max_j \sum_{i=1}^n \phi_i^\top \psi_j.$$

The proof is completed. □

**Lemma 6.** *Let $\{\phi_i\}_{i=1}^n$ be a group of vectors in $\mathbb{R}^d$ such that $\|\phi_i\|_2 \leq L$. Fix $\lambda > 0$ and let $\Lambda_i = \lambda \mathbf{I} + \sum_{i'=1}^i \phi_i \phi_i^\top$. For any sequence $0 = i_1 < i_2 < \ldots < i_k = n$,*

$$\sum_{j=1}^k \min \left\{ \sum_{i=i_j+1}^{i_{j+1}} \phi_i^\top \Lambda_{i_j}^{-1} \phi_i, 1 \right\} \leq 6d \log(nL/\lambda). \tag{18}$$

*Proof.* Let $\mathcal{J} \subset [k-1]$ be the indices $j$ such that $\Lambda_{i_{j+1}} \preccurlyeq 2\Lambda_{i_j}$ does not hold. Then we have

$$2^{|\mathcal{J}|} \leq \Pi_{j \in \mathcal{J}} \frac{\det(\Lambda_{i_{j+1}})}{\det(\Lambda_{i_j})} \leq (nL^2)^d, \tag{19}$$

which implies $|\mathcal{J}| \leq 2d \log_2(nL/\lambda)$.

Continue the computation,

$$\begin{aligned}
\sum_{j=1}^k \min \left\{ \sum_{i=i_j+1}^{i_{j+1}} \phi_i^\top \Lambda_{i_j}^{-1} \phi_i, 1 \right\} &\leq |\mathcal{J}| + \sum_{j \notin \mathcal{J}} \sum_{i=i_j+1}^{i_{j+1}} \phi_i^\top \Lambda_{i_j}^{-1} \phi_i \\
&\leq 2d \log_2(nL/\lambda) + 2 \sum_{i=1}^n \phi_i^\top \Lambda_{i+1}^{-1} \phi_i \\
&\leq 6d \log_2(nL/\lambda). \tag{20}
\end{aligned}$$

The proof is completed. □

## B REGRET ANALYSIS (PROOF OF THEOREM 1)

In this section, we present regret analysis for Algorithm 1, i.e., the proof of Theorem 1.

### B.1 THE SUCCESSFUL EVENT

We first introduce the successful event $\mathcal{G}$. Fix $k \in [K]$. For any $v \in \mathcal{W}_\epsilon$, let $(\hat{\theta}^k(v), \tilde{\theta}^k(v), \Lambda^k(v))$ be the output of Algorithm 2 with input as $\{s_h^{k'}, a_h^{k'}, s_{h+1}^{k'}\}_{k' \in [k-1], h \in [H]}$ and $v$. Recall that $\theta(v) = \mu^\top v$. Let $\kappa = 13\sqrt{6d^2 \log^2(KH/\delta) + 72 \log(KH/\delta)}$. Define $\mathcal{G}^k(v)$ to be the event where

$$\|\theta(v) - \hat{\theta}(v)\|_{\Lambda^k(v)} \leq \kappa, \quad \|\theta(v^2) - \tilde{\theta}(v)\|_{\Lambda^k(v)} \leq 4\kappa. \tag{21}$$

With Lemma 10, we have that $\Pr(\mathcal{G}^k(v)) \geq 1 - 10KH\delta/|\mathcal{W}_\epsilon|$.

Define $\mathcal{G}^k = \cap_{v \in \mathcal{W}_\epsilon} \mathcal{G}^k(v)$ and $\mathcal{G}_\mu = \cap_k \mathcal{G}^k$. Then we have that $\Pr(\mathcal{G}_\mu) \geq 1 - 10K^2H\delta$. On the other hand, we define $\mathcal{G}_r = \{\theta_r \in \Theta^k, \forall 1 \leq k \leq K\}$. By Lemma 13, we have that $\Pr(\mathcal{G}_r) \geq 1 - 10KH\delta$. Let $\mathcal{G} = \mathcal{G}_\mu \cap \mathcal{G}_r$. It then holds that $\Pr(\mathcal{G}) \geq 1 - 20KH\delta$. In the rest of this section, we continue the proof conditioned on $\mathcal{G}$.

## B.2   REGRET DECOMPOSITION

We start with showing that the maintained value function and $Q$-function are nearly optimistic.

**Lemma 7.** *Conditioned on $\mathcal{G}$, it holds that $\theta_r \in \Theta^k$ and $\mu \in \mathcal{U}^k$ for any $k$.*

*Proof.* Recall that

$$\mathcal{U}^k = \{\tilde{\mu} \in \mathcal{U} \mid |\phi^\top \tilde{\mu}^\top v - \phi^\top \hat{\theta}^k(v)| \leq b^k(v, \phi), \forall \phi \in \Phi(\epsilon), v \in \mathcal{W}_\epsilon\}$$

and $b^k(v, \phi) = \alpha\|\phi\|_{(\Lambda^k(v))^{-1}} + 4\epsilon$. By the definition of $\mathcal{G}$, we have that for any $\phi \in \Phi$ and $v \in \mathcal{W}_\epsilon$,

$$|\phi^\top \mu^\top v - \phi^\top \hat{\theta}^k(v)| \leq \|\phi\|_{(\Lambda^k(v))^{-1}} \cdot \|\theta(v) - \hat{\theta}^k(v)\|_{\Lambda^k(v)} \leq \kappa\|\phi\|_{(\Lambda^k(v))^{-1}} \leq b^k(v, \phi).$$

As a result, $\mu \in \mathcal{U}^k$. On the other hand, $\theta_r \in \Theta^k$ by the definition of $\mathcal{G}$. $\square$

For any proper $s, h, k$, we define $V_h^k(s) := \mathbb{E}_{\pi^k}[\sum_{h'=h}^H r_{h'} | s_h = s, \tilde{\mu}^k, \theta^k]$ to be the value function w.r.t. the model $(\mu^k, \theta^k)$.

By definition of regret, we have that

$$\text{Regret(K)}$$

$$= \sum_{k=1}^K \min\left\{\left(V_1^*(s_1^k) - V_1^{\pi^k}(s_1^k)\right), 1\right\}$$

$$\leq \sum_{k=1}^K \min\left\{\left(V_1^k(s_1^k) - V_1^{\pi^k}(s_1^k)\right), 1\right\}$$

$$\leq \sum_{k=1}^K \min\underbrace{\left\{\sum_{h=1}^H \left((\phi_h^k)^\top \theta^k - (\phi_h^k)^\top \theta_r + (\phi_h^k)^\top (\mu^k)^\top V_{h+1}^k - (\phi_h^k)^\top \mu^\top V_{h+1}^k\right), 1\right\}}_{T_1(k)}$$

$$+ \sum_{k=1}^K \min\underbrace{\left\{\sum_{h=1}^H \left((\phi_h^k)^\top \mu^\top V_{h+1}^k - V_{h+1}^k(s_{h+1}^k)\right), 1\right\}}_{T_2(k)}$$

$$+ \sum_{k=1}^K \underbrace{\left(\sum_{h=1}^H r_h^k - V_1^{\pi^k}(s_1^k)\right)}_{T_3(k)} \tag{22}$$

Here the first inequality holds by the optimality of $(\tilde{\mu}^k, \theta^k)$. The right hand side of (22) consists of three terms, where $\sum_k T_1(k)$ is the error due to inaccurate transition and reward model, $\sum_k T_2(k)$ is the martingale difference due to state transition, and $\sum_k T_3(k)$ is the difference between the expected accumulative reward and the empirical accumulative reward. We have the lemma below to bound these terms.

**Lemma 8.** *Conditioned on $\mathcal{G}$, with probability $1 - 10KH\delta$, it holds that*

$$\sum_{k=1}^K T_1(k) \leq \tilde{O}(d^{5.5}\sqrt{K} + d^{6.5}).$$

**Lemma 9.** *Conditioned on $\mathcal{G}$, with probability $1 - 10KH\delta$, it holds that*

$$\sum_{k=1}^K (T_2(k) + T_3(k)) \leq 8\sqrt{K\iota} + 21\iota. \tag{23}$$

The proofs of Lemma 8 and Lemma 9 are presented in Appendix B.4.1 and Appendix B.4.2 respectively.

## B.3 PUTTING ALL PIECES TOGETHER

By (22), Lemma 8 and 9, we conclude that, with probability $1 - 50K^2H^2\delta$, it holds that $\text{Regret}(K) = \tilde{O}(d^{5.5}\sqrt{K} + d^{6.5})$. The proof is completed by replacing $\delta$ with $\frac{\delta}{50K^2H^2}$.

## B.4 MISSING PROOFS

### B.4.1 BOUND OF TERM $T_1(k)$ (PROOF OF LEMMA 8)

Direct computation gives that

$$
\sum_{k=1}^{K} T_1(k) = \sum_{k=1}^{K} \min \left\{ \sum_{h=1}^{H} \left( (\phi_h^k)^\top \theta^k - (\phi_h^k)^\top \theta_r + (\phi_h^k)^\top (\tilde{\mu}^k)^\top V_{h+1}^k - (\phi_h^k)^\top \mu^\top V_{h+1}^k \right), 1 \right\}
$$

$$
\leq \sum_{k=1}^{K} \min \left\{ \sum_{h=1}^{H} ((\phi_h^k)^\top \theta^k - (\phi_h^k)^\top \theta_r), 1 \right\} + \sum_{k=1}^{K} \min \left\{ \sum_{h=1}^{H} ((\phi_h^k)^\top (\tilde{\mu}^k)^\top V_{h+1}^k - (\phi_h^k)^\top \mu^\top V_{h+1}^k), 1 \right\}. \tag{24}
$$

By Lemma 13, with probability $1 - \delta$,

$$
\sum_{k=1}^{K} \min \left\{ \sum_{h=1}^{H} ((\phi_h^k)^\top \theta^k - (\phi_h^k)^\top \theta_r), 1 \right\} = \tilde{O} \left( d^6 \sqrt{\sum_{k=1}^{K} \sum_{h=1}^{H} \text{Var}(R(s_h^k, a_h^k)/\sqrt{d})} + d^{6.5} \right)
$$

$$
= \tilde{O}(d^{5.5}\sqrt{K} + d^{6.5}). \tag{25}
$$

Here we use the fact that $\sum_{h=1}^{H} \text{Var}(R(s_h^k, a_h^k)) \leq \sum_{h=1}^{H} \bar{R}_h^k \leq 1$ with $\bar{R}_h^k$ as the maximal possible value of $R(s_h^k, a_h^k)$.

As for the second term in (24), noting that $V_{h+1}^k, \in \mathcal{W}$ for any proper $k, h$, letting $\bar{V}_{h+1}^k \in \mathcal{W}_\epsilon$ be such that $\|\bar{V}_{h+1}^k - V_{h+1}^k\|_\infty \leq \epsilon$, we have that

$$
\sum_{k=1}^{K} \min \left\{ \sum_{h=1}^{H} ((\phi_h^k)^\top (\tilde{\mu}^k)^\top V_{h+1}^k - (\phi_h^k)^\top \mu^\top V_{h+1}^k), 1 \right\}
$$

$$
\leq \sum_{k=1}^{K} \min \left\{ \sum_{h=1}^{H} ((\phi_h^k)^\top (\tilde{\mu}^k)^\top \bar{V}_{h+1}^k - (\phi_h^k)^\top \mu^\top \bar{V}_{h+1}^k), 1 \right\} + 2KH\epsilon
$$

$$
\leq \sum_{k=1}^{K} \min \left\{ \sum_{h=1}^{H} b^k(\bar{V}_{h+1}^k, \phi_h^k), 1 \right\} + 2KH\epsilon
$$

$$
\leq \alpha \sum_{k=1}^{K} \min \left\{ \sum_{h=1}^{H} \sqrt{(\phi_h^k)^\top (\Lambda^k(\bar{V}_{h+1}^k))^{-1} \phi_h^k}, 1 \right\} + 6KH\epsilon. \tag{26}
$$

Define $\beta_h^k = \sqrt{(\phi_h^k)^\top (\Lambda^k(\bar{V}_{h+1}^k))^{-1} \phi_h^k}$.

For $v \in \mathbb{R}^S$, let $(\hat{\theta}(v), \tilde{\theta}(v), \Lambda^k(v))$ be the output of Algorithm 2 with input as $\{s_{h'}^{k'}, a_{h'}^{k'}, s_{h'+1}^k, \}_{k' \in [k-1], h' \in [H]}$ and $v$. Define

$$
(\sigma_h^k(v))^2 = (\phi_h^k)^\top \tilde{\theta}(v) - \left( (\phi_h^k)^\top \hat{\theta}(v) \right)^2 + 16\alpha\sqrt{(\phi_h^k)^\top (\Lambda^k(v))^{-1} \phi_h^k} + 4\epsilon. \tag{27}
$$

In words, $(\sigma_h^k(v))^2$ is the estimator for the variance $\mathbb{V}(P_{s_h^k, a_h^k}, v)$. By the definition of $\mathcal{G}$ and the fact that $\bar{V}_{h'}^{k'}, (\bar{V}_{h'}^{k'})^2 \in \mathcal{W}_\epsilon$, we have that $\sigma_h^k(\bar{V}_{h'}^{k'}) \geq \mathbb{V}(P_{s_h^k, a_h^k}, \bar{V}_{h'}^{k'})$. Let $i_{\max} = \log_2(H) + 1$. Recall $\mathcal{H}_i = \{h | H - \frac{H}{2^{i-1}} + 1 \leq h \leq H - \frac{H}{2^i}\}$ for $i = 1, 2, \ldots, i_{\max} - 1$ and $\mathcal{H}_{i_{\max}} = \{H\}$. Then we let $\mathcal{V}_i = \{\bar{V}_{h+1}^k | 1 \leq k \leq K, h \in \mathcal{H}_i\}$ for $i = 1, 2, \ldots, \log_2(H)$ and $\mathcal{V}_{i_{\max}} = \{\bar{V}_{H+1}^k | 1 \leq k \leq K\}$.

Fix $i$. For $h \in \mathcal{H}_i$, we define $\Lambda_{(i)}^{\bar{k}}$ as

$$\bar{\Lambda}_{(i)}^k = \lambda \mathbf{I} + \sum_{k'=1}^{k-1} \sum_{h \in \mathcal{H}_i} \frac{\phi_{h'}^{k'}(\phi_{h'}^{k'})^\top}{\max_{v \in \mathcal{V}_i}(\sigma_h^{k'}(v))^2}. \tag{28}$$

By definition, it holds that $\bar{\Lambda}_{(i)}^k \preccurlyeq \Lambda^k(\bar{V}_{h+1}^k)$ for any $k$ and $h' \in \mathcal{H}_i$. It then holds that

$$\sum_{k=1}^{K} \sum_{i=1}^{i_{\max}} \min \left\{ \sum_{h \in \mathcal{H}_i} \beta_h^k, 1 \right\}$$
$$= \sum_{k=1}^{K} \sum_{i=1}^{i_{\max}} \min \left\{ \sum_{h \in \mathcal{H}_i} \sqrt{(\phi_h^k)^\top (\Lambda^k(\bar{V}_{h+1}^k))^{-1} \phi_h^k}, 1 \right\}. \tag{29}$$

Noting that $\bar{\Lambda}_{(i)}^k + \sum_{h \in \mathcal{H}_i} \frac{\phi_h^k (\phi_h^k)^\top}{\max_{v \in \mathcal{V}_i}(\sigma_h^k(v))^2} = \bar{\Lambda}_{(i)}^{k+1}$, by Lemma 6, we have that

$$\sum_{k=1}^{K} \min \left\{ \sum_{h \in \mathcal{H}_i} \frac{(\phi_h^k)^\top (\bar{\Lambda}_{(i)}^k)^{-1} \phi_h^k}{\max_{v \in \mathcal{V}_i}(\sigma_h^k(v))^2}, 1 \right\} \leq 16d \log(KH), \tag{30}$$

$$\sum_{k=1}^{K} I_i^k \leq 16d \log(KH). \tag{31}$$

where $I_i^k = \mathbb{I}\left[ \sum_{h \in \mathcal{H}_i} \frac{(\phi_h^k)^\top (\bar{\Lambda}_{(i)}^k)^{-1} \phi_h^k}{\max_{v \in \mathcal{V}_i}(\sigma_h^k(v))^2} > 1 \right]$.

By Lemma 6, we have

$$\sum_{k=1}^{K} (1 - I_i^k) \sum_{h \in \mathcal{H}_i} \sqrt{(\phi_h^k)^\top (\bar{\Lambda}_{(i)}^k)^{-1} \phi_h^k} \leq \sqrt{\sum_{k=1}^{K} \sum_{h \in \mathcal{H}_i} \frac{(\phi_h^k)^\top (\bar{\Lambda}_{(i)}^k)^{-1} \phi_h^k}{\max_{v \in \mathcal{V}_i}(\sigma_h^k(v))^2}} \cdot \sqrt{\sum_{k=1}^{K} (1 - I_i^k) \sum_{h \in \mathcal{H}_i} \max_{v \in \mathcal{V}_i}(\sigma_h^k(v))^2}$$

$$\leq 16d \log(KH) \cdot \sqrt{\sum_{k=1}^{K} (1 - I_i^k) \sum_{h \in \mathcal{H}_i} \max_{v \in \mathcal{V}_i}(\sigma_h^k(v))^2}. \tag{32}$$

By the successful event $\mathcal{G}$, and noting that $v \in \mathcal{W}_\epsilon$, we have that

$$(\sigma_h^k(v))^2 \leq \mathbb{V}(P_{s_h^k, a_h^k}, v) + (\phi_h^k)^\top (\tilde{\theta}(v) - \theta(v^2)) - 2|(\phi_h^k)^\top (\hat{\theta}(v) - \theta(v))| + 16\alpha \sqrt{(\phi_h^k)^\top (\Lambda^k(v))^{-1} \phi_h^k} + 4\epsilon$$

$$\leq \mathbb{V}(P_{s_h^k, a_h^k}, v) + (6\kappa + 16\alpha) \sqrt{(\phi_h^k)^\top (\Lambda^k(v))^{-1} \phi_h^k} + \frac{1}{KH} \tag{33}$$

$$\leq \mathbb{V}(P_{s_h^k, a_h^k}, v) + (6\kappa + 16\alpha) \sqrt{(\phi_h^k)^\top (\bar{\Lambda}^k)^{-1} \phi_h^k} + \frac{1}{KH}$$

$$\leq \mathbb{V}(P_{s_h^k, a_h^k}, v) + 32\alpha \sqrt{(\phi_h^k)^\top (\bar{\Lambda}^k)^{-1} \phi_h^k} + \frac{1}{KH}.$$

Therefore,

$$\sum_{k=1}^{K} (1 - I_i^k) \sum_{h \in \mathcal{H}_i} \max_{v \in \mathcal{V}_i}(\sigma_h^k(v))^2 \leq \sum_{k=1}^{K} \sum_{h \in \mathcal{H}_i} \max_{v \in \mathcal{V}_i} \mathbb{V}(P_{s_h^k, a_h^k}, v) + 32\alpha \sum_{k=1}^{K} (1 - I_i^k) \sum_{h \in \mathcal{H}_i} \sqrt{(\phi_h^k)^\top (\bar{\Lambda}_{(i)}^k)^{-1} \phi_h^k} + 1. \tag{34}$$

By (32) and (34), we obtain that

$$\sum_{k=1}^{K} (1 - I_i^k) \sum_{h \in \mathcal{H}_i} \sqrt{(\phi_h^k)^\top (\bar{\Lambda}_{(i)}^k)^{-1} \phi_h^k}$$

$$\leq 16d \log(KH) \cdot \sqrt{\sum_{k=1}^{K} \sum_{h \in \mathcal{H}_i} \max_{v \in \mathcal{V}_i} \mathbb{V}(P_{s_h^k, a_h^k}, v) + 32\alpha \sum_{k=1}^{K} (1 - I_i^k) \sum_{h \in \mathcal{H}_i} \sqrt{(\phi_h^k)^\top (\bar{\Lambda}_{(i)}^k)^{-1} \phi_h^k} + 1}, \tag{35}$$

which implies that

$$\sum_{k=1}^{K}(1-I_i^k)\sum_{h\in\mathcal{H}_i}\sqrt{(\phi_h^k)^\top(\bar{\Lambda}_{(i)}^k)^{-1}\phi_h^k} \leq 32\log(KH)\sqrt{\sum_{k=1}^{K}\sum_{h\in\mathcal{H}_i}\max_{v\in\mathcal{V}_i}\mathbb{V}(P_{s_h^k,a_h^k},v)+1}+25000\log^2(KH)\alpha.$$
$$(36)$$

Using (29), (30), (31) and (36), we learn that

$$\sum_{k=1}^{K}\sum_{i=1}^{i_{\max}}\min\left\{\sum_{h\in\mathcal{H}_i}\beta_h^k,1\right\}$$
$$=\sum_{k=1}^{K}\sum_{i=1}^{i_{\max}}\min\left\{\sum_{h\in\mathcal{H}_i}\sqrt{(\phi_h^k)^\top(\Lambda^k(\bar{V}_{h+1}^k))^{-1}\phi_h^k},1\right\}$$
$$\leq \sum_{i=1}^{i_{\max}}\sum_{k=1}^{K}(1-I_i^k)\sum_{h\in\mathcal{H}_i}\sqrt{(\phi_h^k)^\top(\bar{\Lambda}_{(i)}^k)^{-1}\phi_h^k}+\sum_{i=1}^{i_{\max}}\sum_{k=1}^{K}I_i^k \qquad (37)$$
$$\leq i_{\max}\gamma+\sum_{i=1}^{i_{\max}}\left(2\gamma\sqrt{\sum_{k=1}^{K}\sum_{h\in\mathcal{H}_i}\max_{v\in\mathcal{V}_i}\mathbb{V}(P_{s_h^k,a_h^k},v)+1}+100\gamma^2\alpha+16KH\epsilon\right) \qquad (38)$$

with $\gamma=16d\log(KH)$.

Below we fix $i$ and bound the total variance term $\sum_{k=1}^{K}\sum_{h\in\mathcal{H}_i}\max_{v\in\mathcal{V}_i}\mathbb{V}(P_{s_h^k,a_h^k},v)$. Let $v\in\mathcal{V}_i$ be fixed.

Using Lemma 11, we have that: for any $v\in\mathcal{V}_i$

$$\sum_{k=1}^{K}\sum_{h\in\mathcal{H}_i}\mathbb{V}(P_{s_h^k,a_h^k},v)\leq 36Kd^2\log(1/\epsilon)\iota, \qquad (39)$$

Let $\gamma_1=36d^2\log(1/\epsilon)\iota$. Noting that

$$\mathbb{V}(P_{s_h^k,a_h^k},v)=(\phi_h^k)^\top\mu^\top v^2-\left((\phi_h^k)^\top\mu^\top v\right)^2=(\phi_h^k)^\top\theta(v^2)-((\phi_h^k)^\top\theta(v))^2,$$

we have that for any $v\in\mathcal{V}_i$,

$$\sum_{k=1}^{K}\sum_{h\in\mathcal{H}_i}\left((\phi_h^k)^\top\theta(v^2)-((\phi_h^k)^\top\theta(v))^2\right)=\sum_{k=1}^{K}\sum_{h\in\mathcal{H}_i}\mathbb{V}(P_{s_h^k,a_h^k},v)\leq K\gamma_1. \qquad (40)$$

By regarding $\begin{bmatrix}\phi_h^k(\phi_h^k)^\top & \phi_h^k \\ (\phi_h^k)^\top & 1\end{bmatrix}_{k\in[K],h\in\mathcal{H}_i}$ and $\begin{bmatrix}-\theta(v)\theta(v)^\top & \frac{1}{2}\theta(v^2) \\ \frac{1}{2}\theta(v^2)^\top & 0\end{bmatrix}_{v\in\mathcal{V}_i}$ as two groups of vectors with dimension $(d+1)^2$ and applying Lemma 5, we obtain that

$$\sum_{k=1}^{K}\sum_{h\in\mathcal{H}_i}\max_{v\in\mathcal{V}_i}\mathbb{V}(P_{s_h^k,a_h^k},v)$$
$$=\sum_{k=1}^{K}\sum_{h\in\mathcal{H}_i}\max_{v\in\mathcal{V}_i}\left((\phi_h^k)^\top\theta(v^2)-((\phi_h^k)^\top\theta(v))^2\right)$$
$$\leq 2(d+1)^2\max_{v\in\mathcal{V}_i}\sum_{k=1}^{K}\sum_{h\in\mathcal{H}_i}\left((\phi_h^k)^\top\theta(v^2)-((\phi_h^k)^\top\theta(v))^2\right)$$
$$\leq 2(d+1)^2K\gamma_1. \qquad (41)$$

By (38) and (41), we obtain that

$$\sum_{k=1}^{K} \min \left\{ \sum_{h \in \mathcal{H}_i} \beta_h^k, 1 \right\} = \tilde{O}(\gamma \sqrt{d^2 \gamma_1 K}) = \tilde{O}(d^3 \sqrt{K}). \tag{42}$$

Taking sum over $i$, putting (24), (25), (26) and (42) together, and noting that $\epsilon = \frac{1}{K^4 H^4}$, with probability $1 - 10KH\delta$, it holds that

$$\sum_{k=1}^{K} T_1(k) \leq \tilde{O}(d^{5.5} \sqrt{K} + d^4 \sqrt{K} + d^{6.5}) = \tilde{O}(d^{5.5} \sqrt{K} + d^{6.5}). \tag{43}$$

### B.4.2 Bound of Term $T_2(k) + T_3(k)$ (Proof of Lemma 9)

With a slight abuse of notation, here we use $pv$ as shorthand of $p^\top v$ for $p \in \Delta^S$ and $v \in \mathbb{R}^S$.

Recall that

$$T_2(k) := \min \left\{ \sum_{h=1}^{H} \left( (\phi_h^k)^\top \mu^\top V_{h+1}^k - V_{h+1}^k(s_{h+1}^k) \right), 1 \right\}$$

$$= \min \left\{ \sum_{h=1}^{H} \left( (P_{s_h^k, a_h^k} V_{h+1}^k - V_{h+1}^k(s_{h+1}^k) \right), 1 \right\}. \tag{44}$$

Using Lemma 3, with probability $1 - 4Kh\delta$, it holds that

$$\sum_{k=1}^{K} T_2(k) \leq 2\sqrt{2} \sqrt{\sum_{k=1}^{K} \sum_{h=1}^{H} \mathbb{V}(P_{s_h^k, a_h^k}, V_{h+1}^k) \iota + 3\iota}$$

$$\leq 2\sqrt{2} \sqrt{\sum_{k=1}^{K} \sum_{h=1}^{H} \mathbb{V}(P_{s_h^k, a_h^k}, V_{h+1}^k) \iota} + 3\iota.$$

Using Lemma 3 and Lemma 2, with probability $1 - 4KH\delta$,

$$\sum_{k=1}^{K} \sum_{h=1}^{H} \mathbb{V}(P_{s_h^k, a_h^k}, V_{h+1}^k) = \sum_{k=1}^{K} \sum_{h=1}^{H} (P_{s_h^k, a_h^k}(V_{h+1}^k)^2 - (P_{s_h^k, a_h^k} V_{h+1}^k)^2)$$

$$\leq \sum_{k=1}^{K} \sum_{h=1}^{H} ((V_h^k)^2(s_h^k) - (P_{s_h^k, a_h^k} V_{h+1}^k)^2) + \sum_{k=1}^{K} \sum_{h=1}^{H} (P_{s_h^k, a_h^k} - \mathbf{1}_{s_{h+1}^k})(V_{h+1}^k)^2$$

$$\leq 2K + 4\sqrt{2} \sqrt{\sum_{k=1}^{K} \sum_{h=1}^{H} \mathbb{V}(P_{s_h^k, a_h^k}, V_{h+1}^k) \iota} + 3\iota. \tag{45}$$

Solving the equation above, we have that $\sum_{k=1}^{K} T_2(k) \leq \sqrt{16K + 240\iota} + 3\iota$.

Noting that $\mathbb{E}[\sum_{h=1}^{H} r_h^k] = V_1^{\pi^k}(s_1^k)$ and $0 \leq \sum_{h=1}^{H} r_h^k \leq 1$ for any $k \in [K]$, with probability $1 - 2\delta$, it holds that $\sum_{k=1}^{K} T_3(k) \leq 2\sqrt{2K\iota} + 2\iota$. The proof is completed.

### B.4.3 Additional Lemmas

**Lemma 10.** *Fix $v \in \mathbb{R}^S$ such that $\|v\|_\infty \leq 1$. Define $\theta(v) = \mu_P^\top v$ and $\theta(v^2) = \mu_P^\top v^2$. Fix $k \in [K]$. Let $(\theta, \tilde{\theta}, \Lambda)$ be the output of Algorithm 2 with input as $\{s_{h'}^{k'}, a_{h'}^{k'}, s_{h'+1}^{k'}\}_{h' \in [H], k' \in [k-1]}$ and $v$. Recall*

$$\kappa = 13\sqrt{6d^2 \log^2(KH/\delta)} + 72 \log(KH/\delta) \leq \alpha$$

*For any $0 < \delta < 1$, with probability $1 - 10KH\delta/|\mathcal{W}_\epsilon|$, it holds that*

$$\|\theta(v) - \theta\|_\Lambda \leq \kappa \quad \text{and} \quad \|\theta(v^2) - \tilde{\theta}\|_\Lambda \leq 4\kappa. \tag{46}$$

*Proof of Lemma 10.* For convenience, we regard the sample $\{s_{h'}^{k'}, a_{h'}^{k'}, s_{h'+1}^{k'}\}$ as the $H(k'-1)+h$-th sample and rewrite $x_{h'}^{k'}$ as $x_{H(k'-1)+h}$ where $x$ can be any proper notations.

Recall that $\mathbb{V}(P_{s,a}, v)$ denote the variance of $v(s')$ where $s'$ is the reward function and next state by taking $(s, a)$. Let $\text{var}_i(v)$ be shorthand of $\mathbb{V}(P_{s,a}, v)$. Let $\bar{\text{var}}_i(v)$ denote the variance of $v^2(s')$ by taking state-action $(s_i, a_i)$. By Lemma 2, we have that $\bar{\text{var}}_i(v) \le 4\text{var}_i(v)$.

Let $\{\sigma_i, \Lambda_i, \tilde{b}_i, b_i\}_{i \ge 1}$ be the variables computed in Algorithm 2. We then have the following claim.

**Claim 1.** *If it holds that*

$$\sigma_i^2 \ge \text{var}_i(v) + 2\alpha \|\phi_i\|_{\Lambda_{i-1}^{-1}} \tag{47}$$

*for each $i \le (k-1)H$, then (46) holds with probability $1 - 5\delta/|\mathcal{W}_\epsilon|$.*

*Proof of Claim 1.* Fix $1 \le i \le (k-1)H$. Recall the definition of $\Lambda_{i-1}, b_{i-1}, \theta_{i-1}, \tilde{b}_{i-1}$ and $\tilde{\theta}_{i-1}$ in Algorithm 2. Because $\bar{\text{var}}_i(v) \le 4\text{var}_i(v)$, we have that

$$4\sigma_i^2 \ge \bar{\text{var}}_i(v) + 8\alpha \|\phi_i\|_{\Lambda_{i-1}^{-1}} \tag{48}$$

for each $i \le (k-1)H$. Let $\epsilon_i = v(s_{i+1}) - \mathbb{E}[v(s_{i+1})|\mathcal{F}_i]$. Using Lemma 1 with $\{\psi_i, \zeta_i\}_{i \ge 1}$ as $\left\{\frac{\phi_i}{\sigma_i}, \frac{\epsilon_i}{\sigma_i}\right\}_{i=1}^{(k-1)H}$ and parameters as $\sigma^2 = 1, R = H^2, L = H^2, c = 1/(KH)^3, \lambda = 1/H^2$, with probability $1 - 5\delta/|\mathcal{W}_\epsilon|$

$$\|\theta_{i-1} - \theta(v)\|_{\Lambda_{i-1}}$$
$$\le 12\sqrt{6d^2 \log^2(KH/\delta)} + 72d \log(KH/\delta) \max_{1 \le i' \le i} |\epsilon_{i'}/\sigma_{i'}| \cdot \min\{1, \|\phi_{i'}/\sigma_{i'}\|_{\Lambda_{i'-1}^{-1}}\} + \frac{1}{H}$$
$$\le 12\sqrt{6d^2 \log^2(KH/\delta)} + 72d \log(KH/\delta) \max_{1 \le i' \le i} \frac{|\epsilon_{i'}| \|\phi_{i'}\|_{\Lambda_{i'-1}^{-1}}}{\sigma_{i'}^2} + \frac{1}{H}$$
$$\le 12\sqrt{6d^2 \log^2(KH/\delta)} + 72d \frac{\log(KH/\delta)}{\alpha} + \frac{1}{H}$$
$$\le \kappa.$$

Here the second last inequality is by (47).

Let $\bar{\epsilon}_i = v^2(s_{i+1}) - \mathbb{E}[v^2(s_{i+1})|\mathcal{F}_i]$. By setting $\{\psi_i, \zeta_i\}_{i \ge 1}$ as $\left\{\frac{\phi_i}{2\sigma_i}, \frac{\bar{\epsilon}_i}{2\sigma_i}\right\}_{i=1}^{(k-1)H}$, by (48), with probabbility $1 - 5\delta/|\epsilon|$,

$$\|\tilde{\theta}_{i-1} - \theta(v^2)\|_{\Lambda_{i-1}/4}$$
$$\le \left(12\sqrt{12d^2 \log^2(KH/\delta)} + 72d \log(KH/\delta) \max_{1 \le i' \le i} |\bar{\epsilon}_{i'}/(2\sigma_{i'})| \cdot \min\{1, \|\phi_{i'}/(2\sigma_{i'})\|_{4\Lambda_{i'-1}^{-1}}\} + \frac{1}{H}\right)$$
$$\le \left(12\sqrt{12d^2 \log^2(KH/\delta)} + 72d \log(KH/\delta) \max_{1 \le i' \le i} \frac{2\|\phi_{i'}\|_{\Lambda_{i'-1}^{-1}}}{4\sigma_{i'}^2} + \frac{1}{H}\right)$$
$$\le \left(12\sqrt{12d^2 \log^2(KH/\delta)} + 72d \log(KH/\delta)/(2\alpha) + \frac{1}{H}\right)$$
$$\le 2\kappa, \tag{49}$$

which implies that $\|\tilde{\theta}_{i-1} - \theta(v^2)\|_{\Lambda_{i-1}} \le 4\kappa$.

$\square$

So it suffices to prove that (47) holds for any $1 \le i \le (k-1)H$. To prove (47), we use induction on $i = 1, 2, \ldots, n = (k-1)H$. By the update rule in Algorithm 2, we have that

$$\sigma_i^2 = \phi_i^\top \tilde{b}_i - (\phi_i^\top b_i)^2 + 16\alpha \sqrt{\phi_i^\top (\Lambda_{i-1})^{-1} \phi_i} + 4\epsilon. \tag{50}$$

By assuming $\sigma_{i'}^2 \geq \operatorname{var}_{i'}(v) + 2\alpha\|\phi_{i'}\|_{\Lambda_{i'-1}^{-1}}$ holds for $1 \leq i' \leq i-1$, using Claim 1 with samples as $\{s_{i'}, a_{i'}, s_{i'+1}, v(s_{i'+1})\}_{i'=1}^{i-1}$, with probability $1 - 5\delta/|\mathcal{W}_\epsilon|$ it holds that

$$|\phi_i^\top \theta_{i-1} - \phi_i^\top \theta(v)| \leq \|\phi_i\|_{\Lambda_{i-1}^{-1}} \cdot \|\theta_{i-1} - \theta(v)\|_{\Lambda_{i-1}} \leq \kappa\sqrt{\phi_i^\top \Lambda_{i-1}^{-1}\phi_i} \leq \alpha\sqrt{\phi_i^\top \Lambda_{i-1}^{-1}\phi_i}. \quad (51)$$

Note that $\theta(v^2) = \mu_P^\top v^2$. Using Claim 1 again, with probability $1 - 5\delta/|\mathcal{W}_\epsilon|$,

$$|\phi_i^\top \tilde{\theta}_{i-1} - \phi_i^\top \theta(v^2)| \leq 4\alpha\sqrt{\phi_i^\top (\Lambda_{i-1})^{-1}\phi_i}. \quad (52)$$

By noting that $\operatorname{var}_i(v) = \phi_i^\top \tilde{\theta}(v) - (\phi_i^\top \theta(v))^2$, we have that

$$\sigma_i^2 \geq \operatorname{var}_i(v) + 10\alpha\sqrt{\phi_i^\top (\Lambda_{i-1})^{-1}\phi_i}.$$

Also recalling that $\bar{\operatorname{var}}_i(v) \leq 4\operatorname{var}_i(v)$, we have that

$$4\sigma_i^2 \geq \bar{\operatorname{var}}_i(v) + 8\alpha\|\phi_i\|_{\Lambda_{i-1}^{-1}}. \quad (53)$$

The proof is completed. $\qquad\square$

**Lemma 11.** *With probability $1 - 4K^2H^2\delta$, for any $i \in [i_{\max}]$ and $v \in \mathcal{V}_i$, it holds that*

$$\sum_{k=1}^{K} \sum_{h \in \mathcal{H}_i} \mathbb{V}(P_{s_h^k, a_h^k}, v) \leq K(36\iota + 18d + 10\log(KH))$$

*Proof.* With a slight abuse of notation, here we use $pv$ as shorthand of $p^\top v$ for $p \in \Delta^S$ and $v \in \mathbb{R}^S$. Define $\delta' = \delta/|\mathcal{W}_\epsilon|$ and let $\iota' = \log(2/\delta')$. Fix $v \in \mathcal{W}_\epsilon$ and let $\bar{v}(s) = \max\{\max_a P_{s,a}v, v(s)\}$. Fix $1 \leq h_1 \leq h_2 \leq H+1$. With probability $1 - \delta'$, it holds that

$$\sum_{h=h_1}^{h_2} \mathbb{V}(P_{s_h^k, a_h^k}, v) = \sum_{h=h_1}^{h_2} \left(P_{s_h^k, a_h^k}v^2 - (P_{s_h^k, a_h^k}v)^2\right)$$

$$= \sum_{h=h_1}^{h_2} \left(P_{s_h^k, a_h^k}v^2 - v^2(s_{h+1}^k)\right) + \sum_{h=h_1}^{h_2} \left(v^2(s_{h+1}^k) - (P_{s_h^k, a_h^k}v)^2\right)$$

$$\underset{(a)}{\leq} 2\sqrt{\sum_{h=h_1}^{h_2} \mathbb{V}(P_{s_h^k, a_h^k}, v)\iota'} + \sum_{h=h_1}^{h_2} \left(v^2(s_h^k) - (P_{s_h^k, a_h^k}v)^2\right) + 4\iota' + 2$$

$$\underset{(b)}{\leq} 4\sqrt{\sum_{h=h_1}^{h_2} \mathbb{V}(P_{s_h^k, a_h^k}, v)\iota'} + 2\sum_{h=h_1}^{h_2} \max\{v(s_h^k) - P_{s_h^k, a_h^k}v, 0\} + 4\iota' + 2$$

$$\underset{(c)}{\leq} 4\sqrt{\sum_{h=h_1}^{h_2} \mathbb{V}(P_{s_h^k, a_h^k}, v)\iota'} + 2\sum_{h=h_1}^{h_2} \left(\bar{v}(s_h^k) - P_{s_h^k, a_h^k}v\right) + 4\iota' + 2$$

$$\underset{(d)}{\leq} 4\sqrt{\sum_{h=h_1}^{h_2} \mathbb{V}(P_{s_h^k, a_h^k}, v)\iota'} + 2\sum_{h=h_1}^{h_2} \left(v(s_h^k) - P_{s_h^k, a_h^k}v\right) + 4\iota' + 2(h_2 - h_1 + 1)\|\bar{v} - v\|_\infty + 2$$

$$\underset{(e)}{\leq} 4\sqrt{\sum_{h=h_1}^{h_2} \mathbb{V}(P_{s_h^k, a_h^k}, v)\iota'} + 4\sqrt{\sum_{h=h_1}^{h_2} \mathbb{V}(P_{s_h^k, a_h^k}, v)} + 12\iota' + 2(h_2 - h_1 + 1)\|\bar{v} - v\|_\infty + 2$$

$$= 8\sqrt{\sum_{h=h_1}^{h_2} \mathbb{V}(P_{s_h^k, a_h^k}, v)\iota'} + 12\iota' + 2(h_2 - h_1 + 1)\|\bar{v} - v\|_\infty + 2,$$

which further implies that

$$\sum_{h=1}^{H} \mathbb{V}(P_{s_h^k, a_h^k}, v) \leq 36\iota' + 6(h_2 - h_1 + 1)\|\bar{v} - v\|_\infty + 6. \tag{54}$$

Here $(a)$ and $(e)$ hold by Lemma 3, $(b)$ holds by Lemma 2, $(c)$ holds by the fact that $\bar{v}(s) \geq P_{s,a}v$ for any proper $(s, a)$, and $(d)$ holds because $(\bar{v}(s_h^k) - v(s_h^k)) \leq \|\bar{v} - v\|_\infty$ for all proper $(h, k)$.

Now we bound $\|\bar{v} - v\|_\infty$. By definition of $\mathcal{V}_i$, if $v \in \mathcal{V}_i$, there then exists some $k \in [K], h \in \mathcal{H}_i$, such that $\|v - V_{h+1}^k\|_\infty \leq \epsilon$. It then follows that $\bar{v}(s) \leq \max_a P_{s,a} V_{h+1}^k + \epsilon \leq V_h(s) + \epsilon$. Therefore, by Lemma 12, we have that $\|\bar{v} - v\|_\infty \leq \epsilon + \|V_h^k - V_{h+1}^k\|_\infty \leq \epsilon + \frac{2^i}{H}$. By choosing $[h_1, h_2] = \mathcal{H}_i$, we have that with probability $1 - \delta'$

$$\mathbb{I}[v \in \mathcal{V}_i] \sum_{h=h_1}^{h_2} \mathbb{V}(P_{s_h^k, a_h^k}, v) \leq 8\sqrt{\sum_{h=h_1}^{h_2} \mathbb{V}(P_{s_h^k, a_h^k}, v)\iota'} + 12\iota' + 2H\epsilon + 4,$$

which implies

$$\mathbb{I}[v \in \mathcal{V}_i] \sum_{h=1}^{H} \mathbb{V}(P_{s_h^k, a_h^k}, v) \leq 36\iota' + 12 + 6H\epsilon \tag{55}$$

with probability $1 - \delta'$. With a union bound over $\mathcal{W}_\epsilon$ and $i \in [i_{\max}]$, we have that: with probability $1 - 4K^2 H^2 \delta$,

$$\sum_{k=1}^{K} \sum_{h \in \mathcal{H}_i} \mathbb{V}(P_{s_h^k, a_h^k}, v) \leq K(36\iota + 18d + 10\log(KH))$$

for any $i$ and $v \in \mathcal{V}_i$. The proof is completed. □

**Lemma 12.** *For any $k \in [K], h \in [H]$, it holds that $\|V_h^k - V_{h+1}^k\|_\infty \leq \frac{2d}{H-h+1}$.*

*Proof.* Fix $k$. Let $l_h = \|V_h^k - V_{h+1}^k\|_\infty$. Let $\Gamma$ denote the Bellman operator under transition kernel $\mu^k$. Since $\|\Gamma(v_1 - v_2)\|_\infty \leq \|v_1 - v_2\|_\infty$ for any $v_1, v_2 \in \mathbb{R}^S$, $l_h$ in non-decreasing in $h$. So it suffices to bound $\sum_{h=1}^{H} l_h$.

By Lemma 4, for any $s$,

$$\begin{aligned} V_h(s) &- V_{h+1}(s) \\ &\leq \max_{\phi \in \Phi} \phi^\top (\mu^k)^\top (V_{h+1} - V_{h+2}) \\ &\leq 2d\bar{\phi}^\top (\mu^k)^\top (V_{h+1} - V_{h+2}). \end{aligned} \tag{56}$$

It then follows that $\sum_{h=1}^{H} l_h \leq 2d\bar{\phi}^\top (\mu^k)^\top \sum_{h=1}^{H} (V_{h+1} - V_{h+2}) \leq 2d$. The proof is completed.

□

## C    DISCUSSION ABOUT VOFUL

We first introduce the VOFUL estimator in Zhang et al. (2021b). The Algorithm is presented in Algorithm 3. Then we have the lemma to bound the error due to uncertainty of reward parameter.

**Lemma 13.** *Let $\{\phi_i\}_{i=1}^{n}$ be a group of feature vectors in $\mathbb{R}^d$ such that $\|\phi_i\|_2 \leq 1$, and $r_i = \phi_i^\top \theta^* + \epsilon_i \in [-1, 1]$ for some $\theta^* \in \mathbb{R}^d$ with $\|\theta^*\|_2 \leq 1$. Let $\bar{\mathcal{F}}_i$ be the $\sigma$-field of $\{\phi_{i'}, r_{i'}\}_{i' \leq i}$. Assume that $\mathbb{E}[\epsilon_i | \bar{\mathcal{F}}_{i-1}] = 0$ and $\mathbb{E}[\epsilon_i^2 | \bar{\mathcal{F}}_{i-1}] = \sigma_i^2$ for any $i \geq 1$. Let $\{\Theta_i\}_{i=1}^{n}$ to be the confidence region for the true parameter $\theta^*$ in Line 5 Algorithm 3 with input as $\{\phi_i\}_{i=1}^{n}$ and $\{r_i\}_{i=1}^{n}$. It then holds that with probability $1 - 10n\delta$: (i) $\theta_r \in \Theta_i$ for any $i \in [n]$; (ii) For any sequence $0 = i_1 < i_2 < \cdots < i_z = n$, it holds that $\sum_{l=1}^{z-1} \min\left\{\sum_{i=i_l+1}^{i_{l+1}} (\max_{\theta \in \Theta_{i_{l+1}}} \phi_i^\top (\theta - \theta^*), 1\right\} = \tilde{O}(d^{5.5}\sqrt{\sum_{i=1}^{n} \sigma_i^2} + d^{6.5}).$*

*Proof.* Now we proof Lemma 13. Firstly, (i) holds by the lemma below.

**Lemma 14** (Lemma 18 in Zhang et al. (2021b)). *With probability $1 - 3\log(n)\delta$, $\theta^* \in \Theta_i$ for any $i \in [n]$.*

To verify (ii), we define

$$\mathcal{L} := \left\{ l \in [z] | \exists j \in [L_2], u \in \mathcal{B}', \sum_{i=1}^{i_{l+1}} \text{clip}_j^2(\phi_i^\top u) + l_j^2 \geq 4(d+2)^2 \sum_{i=1}^{i_l} \text{clip}_j^2(\phi_i^\top u) + l_j^2 \right\}.$$

With Lemma 15 below, we have that $|\mathcal{L}| \leq O(d\log^3(n))$.

**Lemma 15.** *[Lemma 14 in Zhang et al. (2021b)] Let $f$ be a convex function. Let $\phi_1, \phi_2, \ldots, \phi_t \in \mathcal{B}$ be a sequence of vectors. If there exists a sequence $0 = \tau_0 < \tau_1 < \tau_2 < \ldots < \tau_z = t$ such that for each $1 \leq \zeta \leq z$, there exists $\mu_\zeta \in \mathcal{B}$ such that*

$$\sum_{i=1}^{\tau_\zeta} f(\phi_i \mu_\zeta) + \ell^2 > 4(d+2)^2 \times \left( \sum_{i=1}^{\tau_{\zeta-1}} f(\phi_i \mu_\zeta) + \ell^2 \right), \tag{57}$$

*then $z \leq O(d\log^2(dt/\ell))$.*

Then we have that

$$\sum_{l=1}^{z-1} \min \left\{ \sum_{i=i_l+1}^{i_{l+1}} (\max_{\theta \in \Theta_{i_l+1}} \phi_i^\top(\theta - \theta^*), 1 \right\}$$

$$\leq \sum_{l \notin \mathcal{L}} \sum_{i=i_l+1}^{i_{l+1}} \max_{\theta \in \Theta_{i_l+1}} (\phi_i)^\top(\theta - \theta^*) + O(d\log^3(n))$$

$$\leq (d+2) \sum_{l \notin \mathcal{L}} \sum_{i=i_l+1}^{i_{l+1}} \max_{\theta \in \Theta_{i_{l+1}+1}} (\phi_i)^\top(\theta - \theta^*) + O(d\log^3(n)) \tag{58}$$

$$\leq (d+2) \sum_{i=1}^{n} \max_{\theta \in \Theta_i} (\phi_i)^\top(\theta - \theta^*) + O(d\log^3(n)) \tag{59}$$

$$\leq \tilde{O}\left( d^{5.5} \sqrt{\sum_{i=1}^{n} \sigma_i^2} + d^6 \right).$$

Here (58) is by the definition of $\mathcal{L}$, and the last inequality holds because Lemma 19 in Zhang et al. (2021b), which states that

**Lemma 16.** *[Lemma 19 in Zhang et al. (2021b)] Recall the definition of $\{\Theta_i\}_{i=1}^n$ in Algorithm 3. With probability $1 - 5\log(n)\delta$, it holds that*

$$\sum_{i=1}^{n} \max_{\theta \in \Theta_i} \phi_i^\top(\theta - \theta^*) = \tilde{O}\left( d^{4.5} \sqrt{\sum_{i=1}^{n} \sigma_i^2} + d^5 \right).$$

The proof is completed.

$\square$

---

**Algorithm 3** VOFUL: **V**ariance-Aware **O**ptimism in the **F**ace of **U**ncertainty for **L**inear Bandits

---

1: **Input :** $\{\phi_k\}_{k=1}^n, \{r_i\}_{k=1}^n$
2: **Initialize:** $L_2 = \lceil \log_2 n \rceil$, $\ell_j = 2^{2-j} \forall 1 \leq j \leq L_2 + 1$, $\iota = 16d \ln \frac{dn}{\delta}$, $\Lambda_2 = \{1, 2, \ldots, L_2 + 1\}$,
   $\Theta_1 = \mathcal{B}(2)$, Let $\mathcal{B}'$ be an $n^{-3}$-net of $\mathcal{B}$ with size not larger than $(\frac{4}{n})^{3d}$
3: **Construct Confidence Set:**
4: For each $\theta \in \mathcal{B}$, define $\epsilon_k(\theta) = r_k - (\phi_k)^\top \theta$, $\eta_k(\theta) = (\epsilon_k(\theta))^2$.
5: Define confidence set $\Theta_{k+1} = \left( \bigcap_{j \in \Lambda_2} \Theta_{k+1}^j \right) \cap \Theta_k$, where

$$
\Theta_{k+1}^j = \left\{ \theta \in \mathcal{B} \; : \; \left| \sum_{v=1}^k \text{clip}_j((\phi_v)^\top \mu) \epsilon_v(\theta) \right| \leq \sqrt{\sum_{v=1}^k \text{clip}_j^2((\phi_v)^\top u) \eta_v(\theta) \iota + \ell_j \iota}, \forall \mu \in \mathcal{B}' \right\}
$$

(60)

and $\text{clip}_j(\cdot) = \text{clip}(\cdot, \ell_j)$, $\text{clip}(u, l) = \min\{|u|, l\} \frac{u}{|u|}$.

---