# OpenReview forum: "Horizon-Free Regret for Linear Markov Decision Processes"
_ICLR.cc/2024/Conference — ICLR 2024 poster_

### Official Review · Reviewer_pF7R · 2023-10-27

**Soundness:** 3 good
**Presentation:** 3 good
**Contribution:** 2 fair
**Rating:** 6
**Confidence:** 3

**Summary:**

This paper introduces an algorithm for linear MDPs that achieves a regret bound independent of the horizon length ($H$). It specifically analyzes regret by grouping indices with the same maximal total variance for the optimal value function, allowing value functions within the same group, even if values are inhomogeneous, to share samples. Despite its computational inefficiency, this work addresses the research question of whether it's possible to obtain a horizon-independent regret bound in linear MDPs, extending the prior research on horizon-free regret analysis in linear mixture MDPs.

**Strengths:**

In a linear MDP (simply assuming a known reward function), by setting $\theta^\*\_h = \mu^\top V^*\_{h+1}$ for the transition kernel parameter $\mu$, since the action value for each state-action pair is given by $Q^*_h(s,a) = r(s,a) + \phi(s,a)^\top \theta^*_h$, hence finding the optimal action at h-step is the same as solving a linear bandit problem. Based on this perspective it seems like an interesting approach to group $H$ bandit problems and share samples to get a $H$-independent regret bound.

**Weaknesses:**

- While the analysis of horizon-free regret in linear MDPs, as an extension of the algorithm for linear mixture MDPs, is intriguing, the technical novelty in this paper seems somewhat limited. As mentioned in related work, Zhang et al. (2021b) initially proposed a horizon-free algorithm for linear mixture MDPs, and Zhou & Gu (2022) introduced a computationally efficient horizon-free algorithm. Although it is acknowledged that Linear MDPs become fundamentally more challenging as the dimension of unknown parameters grows with the increasing state space, given that the order of regret bound for minimax algorithms with $H$ dependence in both linear MDPs and linear mixture MDPs is the same (Hu et al., 2023), it is regrettable that this paper only presents a computationally inefficient algorithm despite the existence of computationally efficient $H$-free regret bounds in linear mixture MDPs.

(Hu et al., “Nearly minimax optimal reinforcement learning with linear function approximation”, ICML 2022)

- Providing a slightly clearer explanation of the method in the main text would be beneficial. For instance, as mentioned in Technique 3, calculating an upper bound for $\max\_{h' \in [H]} \sum\_{k=1}^K \sum\_{h=1}^H \mathbb{V} (P\_{s^k\_h, a^k\_h}, V^*\_{h'+1})$ is required, but it was stated that a bound for $ \sum\_{k=1}^K \sum\_{i=1}^{\log\_2 H +1} \max\_{h' \in [H\_i]} \sum\_{h \in H\_i} \mathbb{V} (P\_{s^k\_h, a^k\_h}, V^*\_{h+1})$ was given. A more explicit explanation of how these two bounds are connected would enhance clarity.

**Questions:**

- While the set of all possible features $\Phi$ is not precisely defined, if we define it as $\Phi:=  \\{ \phi(s,a) : (s,a) \in \mathcal{S} \times \mathcal{A} \\}$, then the convexity of $\Phi$ implies that for all $\phi_1, \phi_2 \in \Phi$ and \lambda \in [0,1], $\lambda \phi_1 + (1 - \lambda) \phi_2 \in \Phi$. This appears to be a highly restrictive assumption. It would be beneficial to explicitly mention this assumption as a Key assumption like Assumption 1-3 if it is essential, especially considering it is not utilized in existing linear MDP literature (e.g., Jin et al. 2020b).

- On page 7, could you please provide an explanation for the statement, "$\mathbb{V}(P_{s,a}, v) = \phi(s,a)^\top (\theta(v^2)) - (\phi(s,a)^\top \theta(v))^2$" being described as a linear function of the matrix ?

- Could you clarify how the last inequality in equation (11) holds?

- Typo?: It seems that while $l_h^*$ is defined below equation (10), $l_h$ is not defined. Please provide a definition for $l_h$.

- What is the meaning of "size of transition model"?

---

> ### Author Response · Authors · 2023-11-16
> **Response to Reviewer pF7R**
>
> Thanks for your detailed review! Please find our discussions on the computational complexity and the significance of in the common response. For other questions, please find our response below.
>
>
>
>
> **About connection between** $\sum_{k=1}^K \sum_{i=1}^{\log_2(H)+1}\max_{h'\in H_i}\sum_{h\in H_i}\mathbb{V}(P_{s_h^k,a_h^k},V_{h'+1}^*)$ and  $\max_{h'\in [H]}\sum_{k=1}^K \sum_{h=1}^H \mathbb{V}(P_{s_h^k,a_h^k},V_{h'+1}^*)$ (Here we use $H_i$ ($V_i$) to replace $\mathcal{H}_i$ $(\mathcal{V}_i)$ so that the Latex editor works):
>
> We try to give a more detailed explanation as following: Our initial target is to bound
> $$\sum_{k=1}^K \min\left\\{  \sum_{h=1}^H \sqrt{  (\phi_h^k)^{\top}  (\Lambda^k(V_{h+1}^*))^{-1}\phi_h^k  }  ,1\right\\}.$$
>
> Following the naive maximal variance approach, we need to bound the later term  $\max_{h'\in [H]}\sum_{k=1}^K \sum_{h=1}^H \mathbb{V}(P_{s_h^k,a_h^k},V_{h'+1}^*)$, which is still hard (as presented by the counter example).
> Instead, we turn to bound $$\max_{h'\in [H]}\sum_{k=1}^K \sum_{h=1}^H \mathbb{V}(P_{s_h^k,a_h^k},V_{h'+1}^*)\leq \sum_{i=1}^{\log_2(H)+1}\sum_{k=1}^K \min\left\\{ \sum_{h\in H_i}\sqrt{  (\phi_h^k)^{\top}(\Lambda^k(V_{h+1}^*))^{-1}\phi_h^k } ,1\right\\},$$
> where we could use the bound for the first term  $\sum_{k=1}^K \sum_{i=1}^{\log_2(H)+1}\max_{h'\in H_i}\sum_{h\in H_i}\mathbb{V}(P_{s_h^k,a_h^k},V_{h'+1}^*)$.
>
>
>
>
> **About convex assumption of feature space:**
>
> Indeed, for any feature space $\Phi$, we can perform any feature in the convex hull of $\Phi$.  Suppose there are two actions $a_1$ and $a_2$ with feature $\phi_1$ and $\phi_2$ respectively, then we can create a new (virtual) action with feature $p \phi_1+(1-p)\phi_2$ by playing $a_1$ with probability $p$ and $a_2$ with probability $1-p$. Therefore this assumption is without the loss of generality.
>
> **About the statement** "$\mathbb{V}(P_{s,a},v)= \phi(s,a)^{\top} (\theta(v^2)) - (\phi(s,a)^{\top}\theta(v))^2$" in page 7:
>
> We can write $\mathbb{V}(P_{s,a},v)= \phi(s,a)^{\top} (\theta(v^2)) - (\phi(s,a)^{\top}\theta(v))^2$ as $\mathrm{Trace}(L^{\top}(s,a) B(v))$ where
> $$
> L(s,a)=
> \left[\begin{array}{cc}
>   \phi(s,a)\phi^{\top}(s,a)  , &  \phi(s,a),\\\\
>    \phi^{\top}(s,a), & 1
> \end{array}\right] $$
> and
> $$
> B(v)= \left[\begin{array}{cc}
>   -\theta(v)\theta^{\top}(v) ,  &   \theta(v^2)/2,\\\\
>   \theta^{\top}(v^2)/2 ,& 0
> \end{array}\right]
>  $$
> For fixed $(s,a)$, $\mathbb{V}(P_{s,a},v)$ is a linear function of the matrix $B(v)$. In Equation.(42), with Lemma 5 we derive that
> \begin{align}\sum_{k=1}^K\sum_{h\in H_i}\max_{v\in V_i}\mathbb{V}(P_{s_h^k,a_h^k},v) & = \sum_{k=1}^K\sum_{h\in H_i}\max_{v\in V_i}\mathrm{Trace}(L^{\top}(s_h^k,a_h^k)B(v))
> \\\\ & \leq 2(d+1)^2\max_{v\in V_i}\sum_{k=1}^K\sum_{h\in H_i}\mathrm{Trace}(L^{\top}(s_h^k,a_h^k)B(v))\\\\&=2(d+1)^2\max_{v\in V_i}\sum_{k=1}^K\sum_{h\in H_i}\mathbb{V}(P_{s_h^k,a_h^k},v).
> \end{align}
>
>
> **About the last inequality in Eq.(11):**
>
> We have that
> $$ \max_{\phi\in\Phi} 2d\phi^{\top}\sum_{h=1}^{H-1}\mu^{\top} (V_{h+1}-V_{h+2})  =2d \max_{\phi\in \Phi}\phi^{\top}\mu^{\top}V_{2} =2d\max_{s,a}\phi_{s,a}^{\top}\mu^{\top}V_2.$$
> Continuing the computation, we have
> $$ \max_{\phi\in\Phi} 2d\phi^{\top}\sum_{h=1}^{H-1}\mu^{\top} (V_{h+1}-V_{h+2}) =2d\max_{s,a}P_{s,a}^{\top}V_{2}\leq 2d\max_{s}V_1(s)\leq 2d.$$
>
>
> **About $l_h^{*}$:**
>
> Thank you for point this out. It should be $l_h$ but not $l_h^*$.
>
> **About “size of transition model”:**
>
> It means the number of parameters needed to fully describe the transition model.

---

> ### Author Response · Authors · 2023-11-22
> **Follow-Up**
>
> Thank you for your time and efforts in reviewing our work. We have provided detailed clarification to address the issues raised in your comments. If our response has addressed your concerns, we would be grateful if you could re-evaluate our work.
>
> If you have any additional questions or comments, we would be happy to have further discussions.
>
> Thanks,
>
> The authors

---

> > ### Comment · Reviewer_pF7R · 2023-11-22
> >
> > I thank the authors for their response. I am keeping my score.

---

### Official Review · Reviewer_p2b6 · 2023-10-28

**Soundness:** 2 fair
**Presentation:** 2 fair
**Contribution:** 3 good
**Rating:** 5
**Confidence:** 3

**Summary:**

This paper considers a regret bound of linear MDP that has a mild dependency on the horizon $H$. The main results are emphasized in the Thm 1 (2nd page). The transition kernel is uniform over timesteps, the Q function is discountless, and the time horizon H is fixed. The challenge here is the optimal action may depend on the remaining time (i.e., should the model explore more or not), which is reflected in the index $h$ in $Q_h$, $V_h$. The process is sequential; before the beginning of each episode k, we declare policy $\pi^k$ to play it and observe rewards, and evaluation is based on online measure (regret). Assumptions are that 1: bounded total rewards, 2: linear transition kernel (dim: $S \times d$) and rewards (dim: $d$) with constant scale per feature. Assumption 3 states that the transition and reward share parameter \theta (dim: $d$).

The contribution of this paper is to show horizon-free regret (the regret of subpolynomial dependence on H) for MDP under assumption 2 (compared with similar results with assumption 3 by Zhou and Gu 2022). The paper proposed an algorithm (Algs 1--3 combined) to show this regret. The algorithm seems to be computationally inefficient given it defines some quantities that depend on v (a real variable with bounded norm). While the contribution of this paper is written in the paper and it seems fine, I do not consider the paper to be well-written and ready for publication. Since I am not convinced of the correctness of the claim, I lean to be negative on the paper, but I do not have a strong position . I have several questions in the corresponding section for the sake of clarity.

**Strengths:**

* The paper considers interesting question whether linear MDP is learnable with horizon-free sample complexity. The paper considers the setting where $\theta_r, \mu$ are unknown, which is surely interesting to the community.
* Sections 1--2 are well-written, except for related work section where many papers are cited without specific connection, but overall, nice.

**Weaknesses:**

* The algorithm is computationally inefficient with respect to $v$, so the algorithm here is more conceptual than practical.
* The structure of the paper can be improved. It seems the structure follows Zhou & Gu (2022), but the demonstration of "technical challenge for 3.5 pages without demonstrating algorithm (the algorithm is introduced after that) does not make much sense. At least, this 3.5 pages does not help the reader to be convinced of the results more than 1-page summary.
* There are many typos, which makes it hard for the reader to be convinced of the results.

**Questions:**

My largest question is about Algorithm 1. If I understand correctly, these algorithms require optimization over continuous space. First, $\mathcal{W}(\epsilon)$ is not defined in this paper; similar quantities such as $\mathcal{W}$ and $\mathcal{W}_\epsilon$ are defined twice, so I assume these are equivalent. Lines 6--9 in Alg1 loops over $v \in \mathcal{W}(\epsilon)$, which, defines quantities $(\hat{\theta}, \tilde{\theta}, \Lambda^k(v), b^k(v, \phi))$ as a function of $v$. $\mathcal{U}^k$ is also optimized on the space of $v$.
* $\tilde{\theta}(v)$ is not used in the line 10 of Alg 1 and after?

On assumption 2: Is assumption (2d) standard? It seems that the number of unknown parameter $\mu \in \mathbb{R}^{\mathcal{S}\times d}$ can be arbitrarily large and it is highly nontrivial to have regret bound that does not depend on the size of $\mathcal{S}$.

On the similarity with the (contextual) linear bandit problem. The linear bandit problem is where $x$ is disclosed (contextual is $x$ is $x(t)$) and reward is $r(t) = x^\top \theta$ where $\theta$ is an unknown quantity. In the MDP of this paper, transition has uncertainty and the uncertainty on the reward $r(t)$ w.r.t. transition kernel is much more involved.

The following are rather minor opinions:
* VOFUL: Is it by any means related to OFUL? If I understand correctly, OFUL (e.g., Abbasi-Yadkori et al. 2011) is an algorithm for online bandit algorithm whereas Alg 3 is for the uniform confidence bound (also referred as self-normalized bound or martingale bound).
* Typos. Many periods are missing, for examples:
> p2: See Section 3 for more details. Due to space limitation, we postpone the full proof of Theorem 1 to Appendix B
> p6: That is, we need to use all the samples along the trajectory...
> p7: Fix some eps>0

* Lines 5, 11, 13 in Alg 1 are comments and can be used some special characters for indicating that, such as $//$

**Details Of Ethics Concerns:**

The paper is algorithmic and no ethics reviews needed.

---

> ### Author Response · Authors · 2023-11-16
> **Response to Reviewer p2b6**
>
> Thanks for your detailed review! Please find our discussion on the computational complexity in the common response. For other questions, please find our response below.
>
>
> **About paper structure:**
> We have adjusted the structure of this paper and presented the algorithm before introducing the techniques.
>
> **About the typos:**
> We have fixed the typos accordingly in the revision. Thanks for the effort to check the correctness of this work.
>
> **About Algorithm 1:**
>
> **About $\mathcal{W}(\epsilon)$:**
>
> $\mathcal{W}$ is  a continuous set and $\mathcal{W}_{\epsilon}$ is the discretization of $\mathcal{W}$.
>
> You are correct that $\mathcal{W}(\epsilon)$ and $\mathcal{W}_{\epsilon}$ denote the same set.
>
> We replaced $\mathcal{W}(\epsilon)$ by $\mathcal{W}_{\epsilon}$ and updated the paper accordingly. Thanks for pointing out!
>
>
>
> **$\tilde{\theta}(v)$ is not used in the line 10 of Alg 1 and after?**
>
> We do not use $\tilde{\theta}(v)$ in the main algorithm.  $\tilde{\theta}$ in the output of Algorithm 3 is used in the statement and proof of Lemma 10, which states that $(\hat{\theta}, \tilde{\theta})$ is a close estimation of $(\theta(v),\theta(v^2) )$.
>
>
> **Assumption 2: Is assumption (2d) standard?**
>
> Assumption (2d) could be find in Section 8.1 in [Agarwal et al, 2019]. The commonly used assumption [Jin et. al., 2020, He et.al., 2022]: $P_{s,a,s} = \phi_{s,a}^{\top}\theta_{s’}$ with $\||\theta_{s’}\||_2 \leq \sqrt{d}$, implies assumption (2d). It is possible to obtain an $|S|$-independent regret bound due to the linear structure since we are only required to learn $\mu^{\top}v$ for certain vectors $v$.
>
> Additional reference:
>
> [Agarwal et al., 2019] Reinforcement learning: Theory and algorithms
>
> [Jin et. al., 2020] Provably efficient reinforcement learning with linear function approximation
>
> [He et. al. 2022]  Nearly Minimax Optimal Reinforcement Learning for Linear Markov Decision Processes.
>
>
>
> **Similarity with contextual linear bandit problem:**
>
> We believe linear MDP has the same intrinsic hardness as contextual linear bandit problem. The transitions are involved but it suffices to learn $\theta^*_h$  for $h \in [H]$ to make decisions.
>
> **About VOFUL:**
>
> VOFUL is an algorithm with variance-awared regret bound for linear bandits. The high-level idea is to construct a confidence region for the unknown parameter $\theta$ and then choose arm optimistically. Both VOFUL and OFUL rely on optimism but the algorithm structures are quite different.

---

> ### Author Response · Authors · 2023-11-22
> **Follow-Up**
>
> Thank you for your time and efforts in reviewing our work. We have provided detailed clarification to address the issues raised in your comments. If our response has addressed your concerns, we would be grateful if you could re-evaluate our work.
>
> If you have any additional questions or comments, we would be happy to have further discussions.
>
> Thanks,
>
> The authors

---

> > ### Comment · Reviewer_p2b6 · 2023-11-22
> > **Thank you for clarification**
> >
> > Overall, I will be keeping my score mainly due to the structure of the paper, but please weigh less on my review. If the other reviewers consider this paper good enough, I do not object.
> >
> > * Structure of the paper
> >
> > I think the paper looks better. Still, the informal proof sketch of 3.5pp is not a good idea. I know these sketches are better than nothing, and in many cases, they help the reader -- we cannot understand a paper without good intuition, and typically, the key points in 25-page proof can be condensed. Meanwhile, these are nothing guaranteed in the sense of math. Making them into key lemmas enables you to highlight the main results without compromising rigorousness.
> >
> > * Assumption 2: Is assumption (2d) standard?
> > * Similarity with contextual linear bandit problem:
> >
> > I understand, thanks. At least your comment is very clear.
> >
> > * VOFUL (minor)
> >
> > I mean, Algorithm 3 builds some confidence ball, but no "selection" component.  Optimism in the Face of Uncertainty (OFUL) implies choosing the argmax of some function in a confidence ball, so Algorithm 3 does not represent its name. I do not think this affects the decision, so just please ignore this.

---

> > > ### Author Response · Authors · 2023-11-23
> > > **Thank you for your suggestions**
> > >
> > > Thank you for your suggestions!
> > > We will further polish our paper structure in the final version, including adding key lemmas in the proof sketch section.

---

### Official Review · Reviewer_anFv · 2023-10-30

**Soundness:** 3 good
**Presentation:** 3 good
**Contribution:** 3 good
**Rating:** 6
**Confidence:** 2

**Summary:**

The paper provides the first horizon-free bound for linear MDP. The corresponding algorithm is also novel since this algorithm estimates the value function and corresponding uncertainty directly instead of estimating the transition probability.

**Strengths:**

1. This is the first bound for the linear MDP problem.

2. The approach is novel and interesting.

**Weaknesses:**

1. It would be better to have some numerical illustrations to see whether the given algorithm has better numerical performances.

**Questions:**

1. I wonder whether this algorithm is numerically efficient.

---

> ### Author Response · Authors · 2023-11-16
> **Response to Reviewer anFv**
>
> Thanks for the review! Please find our discussion on the computational complexity in the common response.

---

> > ### Comment · Reviewer_anFv · 2023-11-22
> >
> > I appreciate the authors' response, and I will keep my rating.

---

### Official Review · Reviewer_nnTm · 2023-11-01

**Soundness:** 4 excellent
**Presentation:** 3 good
**Contribution:** 3 good
**Rating:** 8
**Confidence:** 4

**Summary:**

The authors propose a novel algorithm for reinforcement learning (RL) in linear Markov Decision Processes (MDPs). The key feature of the presented algorithm is horizon-free regret bounds, that means that the regret of the algorithm scales only polylogarithmically with a problem horizon.

**Strengths:**

- The first algorithm for linear MDPs that attains nearly horizon-free regret bounds;
- Very detailed discussion on novelties of the proposed approach;

**Weaknesses:**

- Algorithm is computationally unfeasible: it requires to optimize over the confidence region of all models of linear MDPs, that is unfeasible in MDPs with infinite number of states;
- Proof of Lemma 11 is unclear for me, see section Questions;

**Questions:**

Here I would like to present questions to proofs and additionally indicate misprints that I found. I would like to increase my score if all the clarifications regarding proofs will be given.

Questions regarding Lemma 4:

- Why it holds $f(l-y)/y^{d-1} ≤ f(l-x)/x^{d-1}$?
- Is there any references that will show why (16) hold?

Lemma 11:

- Possible misprint: After an inequality (a) there is missed square over $v$ in variance and also there should be $v^2(s^k_{h+1})$ in the second term (not $v^2(s^k_h)$);
- Definition of $\bar v(s)$ is not clearly written: is it  $\bar v(s) = \max_{a} P_{s,a} v?
- It is not clear why (c) holds because it is not easily understandable why the presented $\bar v$ maximizes $v(s^k_{h+1})$, or I did not understand the definition of $\bar v$.
- Why (d) holds? What is a constant $c$?

Misprints and undefined or confusing notation.

- eq. (8) — V^* without h
- Equations (9) and (13) — missed square in \bar \sigma^k_h in the second multiplier after Cauchy-Schwartz inequality;
- After equation (9) it claims that \sigma^k_h is variance, whereas before $\sigma^k_h$ was a square root of variance, it is very confusing;
- After eq. (10) — $l^*$ in the beginning and after l without *;
- After definition of $W_{\epsilon}$: w.r.t. should be with points not commas;
- Proof of Lemma 5: $O$ is not defined;
- After equation (28): use of Var as variance whereas usually it was denoted by $\mathbb{V}$, and after it confusing claim $\sigma^k_h(v) \geq Var$.
- After equation (29) — no brackets around $i$ in the inequality $\bar \Lambda^k_i \leq \Lambda^k(\bar V^k_{h+1})$.
- After equation (30) — forgotten subscript $(i)$ in the definition of $\bar \Lambda^k$ and it is not clear why $\bar \Lambda^k + \sum_{h=1}^H … = \bar \Lambda^{k+1}$.
- Equation (43) — tilde in not over $O$ but over all the expression;
- Last parargraph of page 21: underfined norm type $\Vert \bar v - v \Vert_i$;
- Lemma 12 — missed norm type in the statement (is it $\ell_2$ norm or $\ell_\infty$ as it used later?).

---

> ### Author Response · Authors · 2023-11-16
> **Response to Reviewer nnTm**
>
> Thanks for your detailed feedback! We have updated our paper accordingly. Please find our response to your questions below.
>
>
> **About Lemma 4:**
>
> We revise the proof of Lemma 4 in the revision. We are sorry that in the previous version we mistakenly define $\bar{\phi} = \int_{\Phi}\phi du(\phi)$. Indeed we need add a regularization factor as $\bar{\phi} = \int_{\Phi}\phi du(\phi) / \int_{\Phi}du(\phi)$ to make sure $\bar{\phi}$ is the geometry center of $\Phi$.
>
> Here we presented the response to your two questions regarding Lemma 4. Because of the symmetric structure of $\mathbb{R}^n$, we can assume that $\psi = [1,0,0,\ldots,0]^{\top}$. Now we try to explain your questions.
>
> #### Why $\frac{f(l-y)}{y^{d-1}}\leq \frac{f(l-x)}{x^{d-1}}$:
> We have added a detailed proof in the appendix page 13 (marked in red).
> At a high-level, $f(x)$ denote the $d-1$-dimensional volume of the slice $\mathcal{V}(x):=\Phi \cap \{\phi_1  = x | \phi\in \mathbb{R}^d \}$ (here $\phi_1$ denotes the first dimension of $\phi$). Note that $\mathcal{V}(l)$ is non-empty, and $\Phi$ is convex and closed. Let $\tilde{\phi}$ be an arbitrary element of $\mathcal{V}(l)$. We then have that $\frac{x}{y}\mathcal{V}(l-y) + \frac{y-x}{y}\tilde{\phi}:= \{ \frac{x}{y}\phi+\frac{y-x}{y}\tilde{\phi}｜ \phi\in \mathcal{V}(l-y) \}\subset \mathcal{V}(l-x)$, which implies the $d-1$-dimensional volume of $\mathcal{V}(l-x)$ is at least $(\frac{x}{y})^{d-1}$ fraction of that of $\mathcal{V}(l-y)$ for $0<x\leq y \leq l$.
>
> **About (16):**
>
> After correcting the definition $\bar{\phi}$,
> \begin{align}
> z &= \frac{\int_{\Phi}\phi^{\top}\psi du(\phi)}{\int_{\phi}du(\phi)} \nonumber
> \\ & = \lim_{\epsilon\to 0}\sum_{i=0}^{\left\lceil l/\epsilon\right\rceil}\frac{\int_{\phi\in \Phi, \phi^{\top}\psi \in [i\epsilon,(i+1)\epsilon) } \phi^{\top}\psi du(\phi) }{\int_{\Phi}du(\phi)}\nonumber
> \\ &  = \lim_{\epsilon\to 0}\sum_{i=0}^{\left\lceil l/\epsilon\right\rceil}\frac{\int_{\phi\in \Phi, \phi^{\top}\psi \in [i\epsilon,(i+1)\epsilon) } i\epsilon du(\phi) }{\int_{\Phi}du(\phi)}\nonumber
> \\ & = \lim_{\epsilon\to 0}\sum_{i=0}^{\left\lceil l/\epsilon\right\rceil}f(i\epsilon)i\epsilon^2\nonumber
> \\ & = \int_{0}^l f(x)xdx. \nonumber
> \end{align}
>
> By re-arrange the inequality, and noting that $int_{0}^{l}f(x)dx = 1$ we have that
> \begin{align}
> \int_{ 0\leq x \leq z}f(x)\cdot (z-x) d(x)= \int_{ z\leq x\leq l} f(x)(x-z)dx, \nonumber
> \end{align}
>
>
>
>
> **About Lemma 11:**
>
> **About Equation (a).**
> We make a re-arrangement as
> $$\sum_{h=h_1}^{h_2} v^2(s^k_{h+1}) \leq  \sum_{h=h_1}^{h_2} v^2(s^k_{h}) + v^2(s_{h_2+1}^k)\leq  \sum_{h=h_1}^{h_2} v^2(s^k_{h})+1$$
>
> **About $\bar{v}(s)$:**
> It should be $\bar{v}(s) = \max\{ \max_a P_{s,a}v, v(s) \}$.
>
> **About Equation \(c\).** with the definition of $\bar{v}(s)$ above, we have
> $$\max\{ v(s_h^k)-P_{s,a}v, 0 \} \leq  \max\{ \bar{v}(s_h^k)-P_{s,a}v,0 \} =\bar{v}(s_h^k)-P_{s,a}v.  $$
>
>
> **About Equation (d).**
> The term c should be replaced by $\|\bar{v}-v\|_{\infty}$. Sorry for this mistake.
>
> **Typos:**
>
> **About $\sigma$:** we use $\sigma^2$ to denote the variance term throughout the paper. We fix the typos in the revision.
>
> **About $O$ in the proof of Lemma 5.** $O$ should be $\max_j \sum_{i=1}^n \phi_i^{\top}\psi_j$.  We fix this inequality and delete $O$.
>
> **About $Var$ in Eq.(28):** we replace it by $\mathbb{V}$.
>
> **About Eq.(29):** We add the bracket to $i$.
>
> **About Eq.(30):** We replace $\bar{\Lambda}^k$ by $\bar{\Lambda}^k_{(i)}$ in the analysis after Eq.(30).
>  The equation
> $$\bar{\Lambda}^k_{(i)}+\sum_{h\in H_i}\frac{\phi_{h}^{k}(\phi_{h}^{k})^{\top}}{\max_{v\in V_i}(\sigma_{h}^{k}(v))^2} \preccurlyeq \bar{\Lambda}^k_{(i)} + \sum_{h=1}^H\frac{\phi_{h}^{k} (\phi_{h}^{k})^{\top}}{\max_{v\in V_i}(\sigma_{h}^{k}(v))^2} =\bar{\Lambda}^{k+1}_{(i)}$$
>
> should be replaced by
> $$\bar{\Lambda}^k_{(i)} + \sum_{h\in H_{i}}\frac{\phi_{h}^{k} (\phi_{h}^{k})^{\top}}{\max_{v\in V_i}(\sigma_{h}^{k}(v))^2}  =\bar{\Lambda}^{k+1}_{(i)}$$
>
>
> **About $\|\bar{v}-v\|_i$:** it should be $\|\bar{v}-v\|_{\infty}$.
>
> **About norm type in Lemma 12:** it should be $\infty$ norm

---

> > ### Comment · Reviewer_nnTm · 2023-11-18
> >
> > Thank you for your detailed response!
> >
> > Now the proofs of Lemma 4 and Lemma 11 are much more clear for me and I am happy to increase my score.

---

### Author Response · Authors · 2023-11-16
**Response to Common Questions:**

We thank the reviewers for valuable feedback. We have carefully fixed the typos in the  revision.  Below we present the response to the major concerns. We also present response to each reviewer for their additional concerns.

**About computation complexity:**

We acknowledge that the computational cost is a major limitation of this work. One major bottleneck is that we need to invoke a variance-aware linear bandit algorithm to learn the reward function, where  it  remains an open problem for computational efficient variance-aware regret bound for linear bandit problem.

We also want to note that the first horizon-free works for tabular MDP [Wang et al. 2020] and linear mixture [Zhang et al. 2021b] are not computationally efficient as well, but greatly inspire follow-up works on computational efficient algorithms with horizon-free regret bounds. Therefore, we believe our work already serves an important step toward deeper theoretical understanding of the popular linear MDP model.

**Significance of this paper:**

We want to emphasize that the existing techniques in optimal bounds for linear MDP [Hu et. al., 2023, He et. al., 2022 ] do not indicate that the horizon-free case is trivial.  All existing algorithms with horizon-free bounds try to fully recover the transition model. As a result, their bounds scale with the size of the transition model. The best one for tabular is $\widetilde{O}(\sqrt{SAK}+S^2A)$ [Zhang et al. 2021A] and the best one for linear mixture MDP is $d\sqrt{K}+d^2$ [Zhou and Gu, 2022].

In linear MDP,  roughly speaking, the existing algorithms try to learn $H$ **independent** linear bandit problems for $Q_h^*$ and $V_h^*$ functions from $h=1,...,H$. In the time-inhomogeenous case, each $h$ is independent, and a polynomial dependency on $H$ is unavoidable. This approach can obtain optimal regret [Agarwal et al. 2022].

In the time-homogeneous case, although the transition model is shared across all $h=1,...,H$, the $Q_h^*$ and $V_h^*$ are still different for different $h$. **The key technical challenge** is to relate  $Q_h^*$ and $V_h^*$ from different $h$ without having a regret that depends on the size of the transition model**. To remove all polynomial factors of $H$, it is necessary to share samples among different $h$ (in contrast, existing papers on linear MDP focused on the inhomogeneous case so that there is no need to share samples). As stated in Section3, although we have the desired exploration bonus for each linear bandit separately following [Zhou and Gu, 2022], there is no standard way to bound the sum of the bonus in equation.(1)  due to inhomogeneous information matrix.  In comparison,  in horizon-free algorithms for  tabular MDP and linear mixture MDP, the samples are shared by estimating the transition model.

---

### Meta-Review · Area_Chair_kqpA · 2023-12-07

**Metareview:**

The paper solves a very natural question: can we provide a regret bound for linear MDPs that is independent of the time horizon? It provides an algorithm which achieves such horizon-independence. The main criticism of this work is the high computational complexity of the proposed method, but this is something that can be investigated and improved in follow up work.

**Justification For Why Not Higher Score:**

Given that the main result here is theoretical without, at the moment, significant practical implications, I think poster is more appropriate than spotlight or oral.

**Justification For Why Not Lower Score:**

The paper makes a solid contribution to the ML literature.

---

### Decision · Program_Chairs · 2024-01-16

Accept (poster)